# TempMe: Video Temporal Token Merging for Efficient Text-Video Retrieval

**Leqi Shen**[1,2,3*]   **Tianxiang Hao**[1,2*]   **Tao He**[5,6]   **Sicheng Zhao**[2†]
**Yifeng Zhang**[4]   **Pengzhang Liu**[4]   **Yongjun Bao**[4]   **Guiguang Ding**[1,2†]
[1] School of Software, Tsinghua University    [2] BNRist, Tsinghua University
[3] Hangzhou Zhuoxi Institute of Brain and Intelligence
[4] JD.com    [5] GRG Banking Equipment Co., Ltd.    [6] South China University of Technology

## Abstract

Most text-video retrieval methods utilize the text-image pre-trained models like CLIP as a backbone. These methods process each sampled frame independently by the image encoder, resulting in high computational overhead and limiting practical deployment. Addressing this, we focus on efficient text-video retrieval by tackling two key challenges: 1. From the perspective of trainable parameters, current parameter-efficient fine-tuning methods incur high inference costs; 2. From the perspective of model complexity, current token compression methods are mainly designed for images to reduce spatial redundancy but overlook temporal redundancy in consecutive frames of a video. To tackle these challenges, we propose Temporal Token Merging (TempMe), a parameter-efficient and training-inference efficient text-video retrieval architecture that minimizes trainable parameters and model complexity. Specifically, we introduce a progressive multi-granularity framework. By gradually combining neighboring clips, we reduce spatio-temporal redundancy and enhance temporal modeling across different frames, leading to improved efficiency and performance. Extensive experiments validate the superiority of our TempMe. Compared to previous parameter-efficient text-video retrieval methods, TempMe achieves superior performance with just **0.50**M trainable parameters. It significantly reduces output tokens by **95**% and GFLOPs by **51**%, while achieving a **1.8**× speedup and a **4.4**% R-Sum improvement. With full fine-tuning, TempMe achieves a significant **7.9**% R-Sum improvement, trains **1.57**× faster, and utilizes **75.2**% GPU memory usage. The code is available at `https://github.com/LunarShen/TempMe`.

## 1 Introduction

In the domain of video-language understanding, text-video retrieval is a crucial task focused on matching videos that correspond to specific query texts or vice versa. Given the powerful capacity of large-scale text-image pre-training Fan et al. (2024); Radford et al. (2021); Xu et al. (2024a), recent studies Luo et al. (2022); Wang et al. (2023); Pei et al. (2023); Jin et al. (2022) have increasingly focused on full fine-tuning of CLIP Radford et al. (2021) for text-video retrieval. Specifically, these methods incorporate cumbersome modules to extract video and text representations. However, the slow inference speed of these methods severely limits their real-world applications. For example, the training process of CLIP4Clip Luo et al. (2022) with CLIP-ViT-B/16 requires 70.1GB GPU memory usage and takes 6.5 hours. Therefore, we focus on efficient fine-tuning text-video retrieval Ju et al. (2022); Huang et al. (2023); Yang et al. (2024) to lower computational expenses, which freeze the pre-trained backbone and introduce minimal trainable parameters to model video data.

Efficient adaptation of text-image pre-trained models for text-video retrieval remains challenging due to the inherent differences between image and video modalities. The shift from processing a single image to handling multiple sampled frames dramatically raises the number of patch tokens fed into the model, complicating the extraction of video representations. As a result, this adaption

---

*Equal contribution. Emails: `lunarshen@gmail.com`, `beyondhtx@gmail.com`
†Corresponding authors. Emails: `schzhao@gmail.com`, `dinggg@tsinghua.edu.cn`

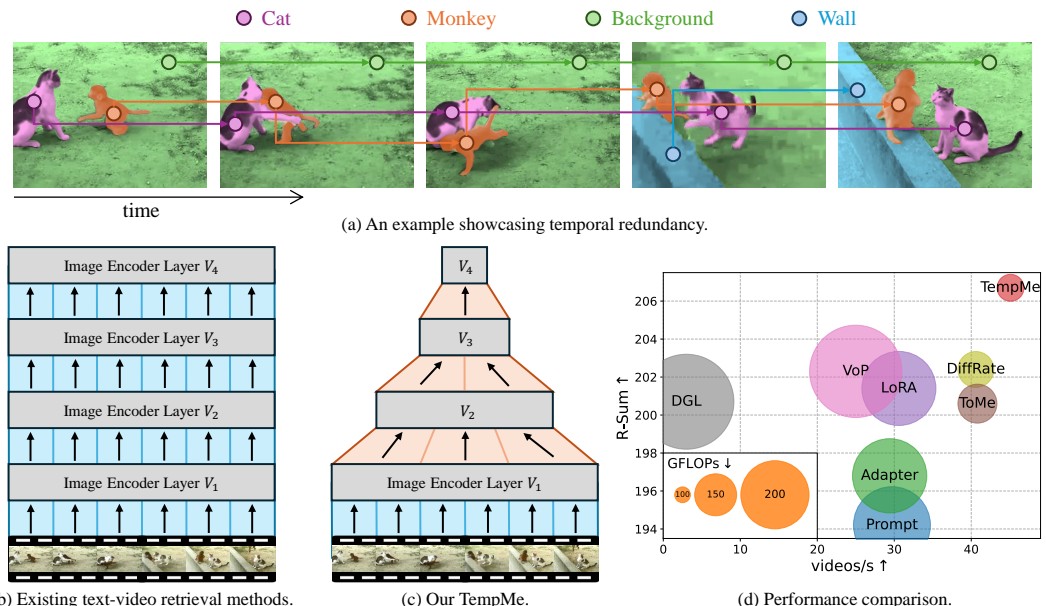

Figure 1: (a) An example illustrates the large temporal redundancy between adjacent frames. Identical subjects are highlighted in the same color. (b) Current methods treat video input as a sequence of multiple sampled frames, causing high complexity due to the large number of tokens. (c) In contrast, our TempMe reduces temporal redundancy by progressively merging redundant tokens in adjacent video clips. (d) With CLIP-ViT-B/16 on MSRVTT, our TempMe reaches state-of-the-art performance with minimal computational overhead. R-Sum is the sum of R@1, R@5, and R@10.

suffers from high computational overhead, which makes practical deployment difficult. As shown in Figure 1b, existing parameter-efficient text-video retrieval methods Huang et al. (2023); Yang et al. (2024); Jin et al. (2024); Cao et al. (2024) treat video input as a sequence of multiple sampled frames, achieving competitive accuracy but at the cost of inference efficiency.

Motivated by these limitations, we argue that *temporal redundancy* plays a major role in the model's high complexity when employing the pre-trained image encoder for video inputs. The sequentially sampled frames in a video capture a continuous temporal progression, illustrating subject evolution, interaction, and transition over time. In Figure 1a, the video shows a cat and a monkey fighting, showing the same subjects in adjacent frames against a repetitive background. This temporal progression introduces substantial redundancy due to repeated information in consecutive frames, contributing to the high complexity resulting from repeated model calculations.

To mitigate computational complexity, a recently popular technique, token compression Rao et al. (2021); Kong et al. (2022); Chen et al. (2023a); Bolya et al. (2022), might be a fascinating solution, which has achieved huge success in accelerating image models. Classically, token compression methods aim to merge redundant spatial tokens in the image to improve efficiency. However, they fail to resolve the unique challenges imposed by video modality. Specifically, they overlook the temporal redundancy across different frames, leading to inferior performance. Currently, there is no dedicated compression algorithm designed for text-video retrieval models, and the efficiency issue urgently needs to be addressed.

In conclusion, existing methods struggle to achieve efficient text-video retrieval: (1) Parameter-efficient methods still lead to high inference costs. (2) Token compression methods fail to address temporal redundancy. In this paper, we propose **Temp**oral Token **Me**rging (TempMe), which effectively integrates parameter-efficient tuning with token compression to overcome these limitations. Particularly, our method adopts a progressive multi-granularity framework, enabling the efficient merging of redundant temporal tokens, as visualized in Figure 1c. Videos can be viewed as aggregations of clips with varying temporal granular levels, from individual images at the micro-level to the entire video at the macro-level. Through gradually combining neighboring clips, the total number of tokens is dramatically decreased, while the features evolve from fine-grained image-level features to holistic video-level features, leading to lower complexity and better performance.

To validate the advantages of our TempMe, extensive experiments are conducted on four benchmark datasets, MSRVTT Xu et al. (2016), ActivityNet Krishna et al. (2017), DiDeMo Anne Hendricks et al. (2017), and LSMDC Rohrbach et al. (2015). Experimental results consistently demonstrate that our TempMe offers a leading balance between speed and accuracy, outperforming other efficient fine-tuning methods. Figure 1d shows the performance comparison of CLIP-ViT-B/16 on MSR-VTT. Compared to VoP Huang et al. (2023) and DGL Yang et al. (2024), TempMe significantly reduces output tokens by $95\%$ and GFLOPs by $51\%$, while achieving a $1.8\times$ speedup and a $4.4\%$ R-Sum improvement. Moreover, TempMe can be seamlessly integrated with various parameter-efficient fine-tuning (see Appendix C.2) and full fine-tuning (see Table 5), which demonstrates its robust generalization capabilities. When fully fine-tuning with CLIP-ViT-B/16, TempMe achieves a significant $7.9\%$ R-Sum improvement, trains $1.57\times$ faster, and utilizes $75.2\%$ GPU memory usage.

Our major contributions are summarized as follows: (1) Our study reveals that temporal redundancy in video content causes excessive computational demands when adapting pre-trained text-image models for text-video retrieval. Existing parameter-efficient methods incur high inference costs, while current token compression methods fail to address temporal redundancy. (2) To overcome such limitations, we propose TempMe, a parameter- and inference-efficient text-video retrieval architecture that merges redundant temporal tokens in adjacent video clips step by step, simplifying model complexity while extracting unified and discriminative video features. (3) Results on four benchmarks reveal that TempMe outperforms current SOTA methods, including both parameter-efficient video-text retrieval and compression methods. Extensive experiments indicate that our TempMe effectively reduces model complexity while achieving superior performance.

## 2 RELATED WORKS

**Text-Video Retrieval.** Due to the great success of language-vision pre-training Wang et al. (2022b); Xue et al. (2022); Chen et al. (2023b); Li et al. (2023); Fan et al. (2024); Radford et al. (2021); Xu et al. (2024a); Li et al. (2021; 2022) in various downstream cross-modal tasks Antol et al. (2015); Xu et al. (2015); Karpathy & Fei-Fei (2015); Shen et al. (2024), numerous studies Luo et al. (2022); Gorti et al. (2022); Liu et al. (2022); Wang et al. (2023); Guan et al. (2023); Jin et al. (2023b); Pei et al. (2023); Jin et al. (2022); Shen et al. (2025); Lei et al. (2021) have achieved remarkable results by utilizing CLIP's Radford et al. (2021) knowledge for text-video retrieval. These approaches typically employ CLIP's encoders as the backbone and use complex similarity calculations. Specifically, CLIP4Clip Luo et al. (2022) introduces several video aggregation schemes to obtain video features. Cap4Video Wu et al. (2023) generates associated captions to improve performance. HBI Jin et al. (2023a) formulates video and text as players in a cooperative game. However, these full fine-tuning methods with their customized modules require significant memory and computational resources. Therefore, our work aims to explore efficient fine-tuning methods to lower computational overhead.

Mainstream parameter-efficient fine-tuning methods in the image domain, such as Prompt Khattak et al. (2023); Jia et al. (2022); Zhou et al. (2022b;a); Hao et al. (2024), LoRA Hu et al. (2021); Zhang et al. (2022); Xiong et al. (2024), and Adapter Houlsby et al. (2019); He et al. (2021); Chen et al. (2022); Xu et al. (2024b); Hao et al. (2023b;a), freeze pre-trained model parameters while adding extra tunable parameters to minimize storage demands. For parameter-efficient text-video retrieval, VoP Huang et al. (2023) and DGL Yang et al. (2024) introduce extra modules to generate prompt tokens and capture global video information. Despite their good performance with small tunable parameters, they face high inference costs due to the significant token number in videos.

**Token Compression** Token compression methods Ren et al. (2023); Liang et al. (2022); Wang et al. (2022a); Long et al. (2023); Rao et al. (2021); Kong et al. (2022); Chen et al. (2023a); Bolya et al. (2022); Marin et al. (2021) aim to reduce computational burden by reducing tokens while preserving the essential information.

Image token compression methods focus on pruning or merging tokens in a single image. DeCo Yao et al. (2024) adopts 2D Adaptive Pooling to downsample vision tokens at the spatial level for MLLMs. EVIT Liang et al. (2022) identifies the attentive tokens and fuses the inattentive tokens. DiffRate Chen et al. (2023a) performs token pruning and merging simultaneously. ToMe Bolya et al. (2022) reduces tokens in a transformer gradually by merging a fixed number of tokens in each block.

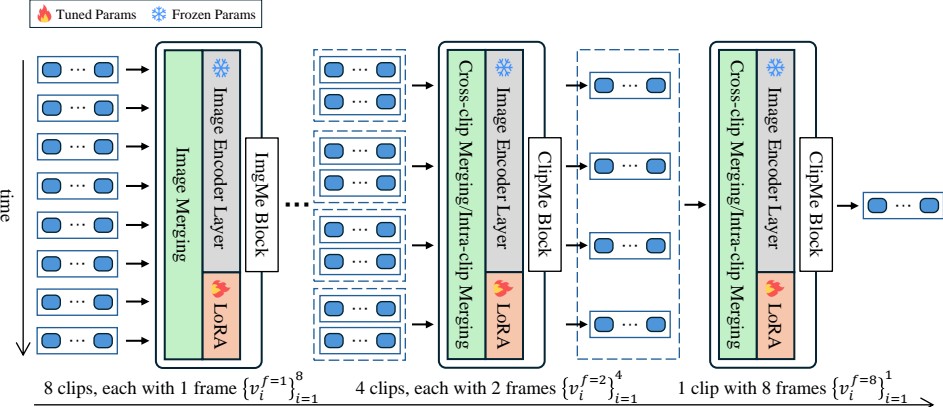

Figure 2: Overview of our proposed TempMe. We introduce a Progressive Multi-Granularity (PMG) framework consisting of both image merging and clip merging stages. In the image merging stage, ImgMe Block merges redundant spatial tokens within a single frame. Following this, ClipMe Block progressively forms new clips from adjacent ones, facilitating video-level feature learning and reducing temporal redundancy by merging tokens across different frames.

Although ToMe has been applied to video classification by simultaneously processing all frame tokens in Spatiotemporal MAE Feichtenhofer et al. (2022), this implementation is incompatible with CLIP's image encoder for text-video retrieval. When processing each sampled frame via CLIP's image encoder, image token compression methods effectively reduce spatial redundancy by merging similar tokens within a single frame. However, they do not address the temporal redundancy across frames, which greatly contributes to computational overhead.

Video token compression methods Wang et al. (2022a); Ding et al. (2023); Ren et al. (2023) focus on pruning or merging tokens across frames. STA Ding et al. (2023) considers temporal redundancy and semantic importance, which is tailored for video architectures where all frame tokens are processed jointly. TESTA Ren et al. (2023) introduces temporal aggregation for video-language pre-training. We re-implement these video token compression methods on CLIP for text-video retrieval.

In this work, we focus on text-video retrieval using CLIP, where each sampled frame is processed as an independent token set. Existing token compression methods are limited to pruning or merging tokens within a single token set for an image or video, without addressing token compression across multiple sets or incorporating temporal fine-tuning. In contrast, we have explored a practical and feasible path to reach both superior performance and computational efficiency. By fruitfully integrating parameter-efficient fine-tuning and token compression techniques, we propose TempMe and reach state-of-the-art performance. TempMe can progressively merge different frame token sets, and thus minimize spatio-temporal redundancy and enhance temporal modeling across frames.

## 3 METHODOLOGY

### 3.1 PRELIMINARIES

Text-video retrieval involves leveraging textual queries to accurately retrieve related videos, as well as utilizing video queries to find associated textual descriptions. The primary goal is to bridge the semantic gap between textual description and video content by learning a similarity function $s(t, v)$. Here, $t$ denotes a text, and $v$ represents a sequence of sampled frame $\{I_i\}_{i=1}^{F}$, where $F$ is the number of sampled frames in time. We employ LoRA Hu et al. (2021) to adapt CLIP Radford et al. (2021) for text-video retrieval tasks. CLIP employs transformer that includes both multi-head self-attention (MHSA) and feed-forward network (FFN). The trainable matrices of LoRA can be merged, introducing no extra inference latency. Further details are provided in Appendix A.

Notably, the classic method, VoP Huang et al. (2023), does not implement specific designs for the text modality. It merely introduces prompts within the text encoder. Similar to VoP, we apply LoRA to the text encoder, while primarily concentrating on the video modality.

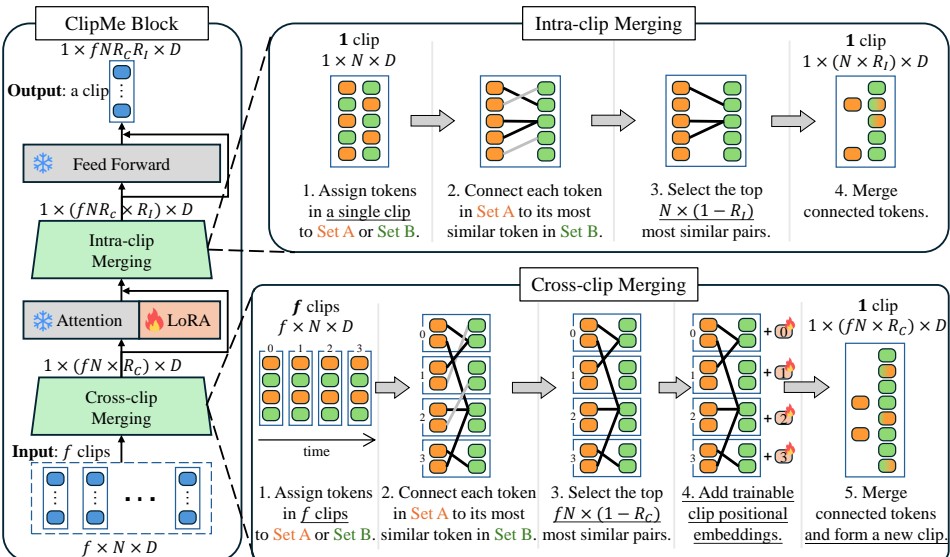

Figure 3: ClipMe Block. Given an input of $f$ clips $\mathbb{R}^{f \times N \times D}$, the cross-clip merging step merges tokens from all clips to form a new clip $\mathbb{R}^{1 \times fNR_c \times D}$. Subsequently, the intra-clip merging step merges tokens within this newly formed clip, producing $\mathbb{R}^{1 \times fNR_cR_I \times D}$. If the input contains only one clip, the cross-clip merging is skipped.

## 3.2 Temporal Token Merging

For efficient text-video retrieval, we freeze the pre-trained CLIP and merely train LoRA in both the image and text encoders. In this subsection, we focus on merging temporal tokens across multiple sampled frames to minimize temporal redundancy and extract more comprehensive video features.

**Progressive Multi-Granularity Framework.** In text-video retrieval, the significant impact of temporal redundancy on model complexity has not been adequately explored. To mitigate temporal redundancy, we proposed the Progressive Multi-Granularity (PMG) framework that efficiently merges similar temporal tokens across different frames. In our PMG framework (see Figure 2), we introduce a two-stage merging approach: image merging and clip merging. Initially, ImgMe block independently encodes each single frame, merging similar tokens within the same frame and leveraging pre-trained image knowledge to capture low-level spatial details. Next, ClipMe block (see Figure 3) is proposed to aggregate short-frame clips into extended-frame clips, merging temporal tokens across different frames and learning unified and discriminative video features.

Specifically, each sampled frame $I$ can be regarded as a discrete video clip lasting only one frame $v^{f=1}$. Concatenating these consecutive clips yields an extended clip with increased temporal length. In the toy example presented in Figure 2, 8 individual frames $\left\{ v_i^{f=1} \right\}_{i=1}^{8}$ are aggregated into four clips, each containing with two frames $\left\{ v_i^{f=2} \right\}_{i=1}^{4}$. These clips are then aggregated to form a complete video clip with all 8 frames $\left\{ v_i^{f=8} \right\}_{i=1}^{1}$. As clips are progressively aggregated, we merge a significant number of similar tokens that arise from overlapping information across frames. Simultaneously, comprehensive video information is captured through the attention mechanism that processes tokens across different frames.

**ImgMe Block.** Inspired by ToMe Bolya et al. (2022), the image merging step merges tokens to reduce by $r$ per layer, which is employed between the MHSA and FFN branches of each ImgMe Block. Following Bipartite Soft Matching Bolya et al. (2022), we first partition the tokens into two sets $\mathbb{A}$ and $\mathbb{B}$ by alternating. For each token in set $\mathbb{A}$, we find its most similar token in Set $\mathbb{B}$ using cosine similarity. The top $r$ most similar pairs are then merged by averaging their features. Given

Table 1: Complexity comparisons. We report the R-Sum metric for 12 sampled frames on MSRVTT and 64 sampled frames on ActivityNet, respectively. The best metric is highlighted in bold.

| Backbone | CLIP-ViT-B/32 | | | | | | CLIP-ViT-B/16 | | |
|---|---|---|---|---|---|---|---|---|---|
| # Frames | | 12 | | | 64 | | | 12 | |
| Method | GFLOPs | # Tokens | R@1/R-Sum | GFLOPs | # Tokens | R@1/R-Sum | GFLOPs | # Tokens | R@1/R-Sum |
| LoRA | 53.0 (100%) | 12 × 50 (100%) | 43.7/193.0 | 276.7 (100%) | 64 × 50 (100%) | 38.7/191.5 | 211.3 (100%) | 12 × 197 (100%) | 47.3/201.4 |
| DiffRate | 36.8 ( 69%) | 12 × 20 ( 40%) | 41.5/189.9 | 190.1 ( 69%) | 64 × 20 ( 40%) | 38.0/188.7 | 138.5 ( 66%) | 12 × 49 ( 25%) | 47.3/202.4 |
| ToMe | 40.2 ( 76%) | 12 × 26 ( 52%) | 42.9/191.4 | 208.5 ( 75%) | 64 × 26 ( 52%) | 38.6/189.6 | 144.4 ( 68%) | 12 × 77 ( 39%) | 46.2/200.6 |
| TempMe | **34.8 ( 65%)** | 1 × 97 ( 16%) | **46.1/198.6** | **180.3 ( 65%)** | 1 × 500 ( 16%) | **44.9/205.6** | 121.4 ( 57%) | 1 × 127 ( 5%) | **49.0/206.7** |

an input $\mathbb{R}^{1 \times N \times D}$, the image merging stage with $K$ ImgMe Block outputs $\mathbb{R}^{1 \times (N-rK) \times D}$, where $N$ denotes the number of tokens, and $D$ denotes the dimension of the token feature.

To prevent loss of image information, $r$ is set to a low value: $r = 2$ for CLIP-ViT-B/32 and $r = 10$ for CLIP-ViT-B/16. However, the model complexity remains high due to the processing of multiple sampled frames. Therefore, we aim to merge temporal tokens to further reduce model complexity.

**ClipMe Block.** Instead of ImgMe Block which merges tokens in each single frame, we propose ClipMe Block to process multi-frame clips as shown in Figure 3. In ClipMe Block, we introduce two novel steps: cross-clip merging and intra-clip merging. During cross-clip merging, adjacent clips are aggregated, which significantly reduces the number of temporal tokens and generates a new clip. The intra-clip merging step further compresses the tokens within the newly formed clip.

In cross-clip merging, we aim to combine tokens with their most similar tokens from the same subject, whether these matches occur within a single frame or span across multiple frames. Initially, all tokens from the input $f$ clips $\mathbb{R}^{f \times N \times D}$ are alternately divided into Set $\mathbb{A}$ or Set $\mathbb{B}$. This alternating assignment scheme is based on the observation that the adjacent tokens often depict the same subject, thus their features tend to be similar. Moreover, this assignment ensures that tokens representing the same subject across various frames are distributed between both sets. Considering the high temporal redundancy observed in videos, we select a large subset, $fN \times (1 - R_c)$, of the most similar token pairs for merging, where $R_C$ denotes the ratio of kept tokens in cross-clip merging. Before merging, the trainable clip positional embeddings are added to facilitate temporal understanding in MHSA. Subsequently, these merged tokens, along with the unmerged tokens, constitute a new clip $\mathbb{R}^{1 \times (fN \times R_c) \times D}$, which is then fed into MHSA.

During intra-clip merging, the tokens in the newly formed clip, improved by MHSA for better temporal and spatial details, are further reduced. In contrast to the image merging step that merges a small number $r$ of tokens in an individual frame, the intra-clip merging step merges a larger proportion of tokens within a clip containing multiple frames, compressing $\mathbb{R}^{1 \times N \times D}$ into $\mathbb{R}^{1 \times (N \times R_I) \times D}$, where $R_I$ denotes the ratio of kept tokens in intra-clip merging.

### 3.3 COMPLEXITY ANALYSIS.

To verify the efficiency of our proposed TempMe, we perform a detailed complexity analysis in Table 1. Following common protocols Luo et al. (2022); Yang et al. (2024), the frame length is set to 12 for MSR-VTT and LSMDC, and to 64 for ActivityNet and DiDeMo. The complexity comparison between ToMe and our TempMe for these frame lengths is shown in Table 1. Compared to ToMe, our TempMe reduces tokens by more than **30**% for both CLIP-ViT-B/32 and CLIP-ViT-B/16, effectively decreasing model complexity while significantly surpassing accuracy. Particularly, with a 12-frame length and CLIP-ViT-B/16 backbone on MSRVTT, TempMe outputs only **5**% of the input tokens, reaches **57**% GFLOPs, and achieves a **5.3**% R-Sum gain.

## 4 EXPERIMENTS

### 4.1 EXPERIMENTAL SETTINGS

**Datasets.** Following common practice, we perform experiments on four widely used benchmarks for text-video retrieval: MSRVTT Xu et al. (2016), ActivityNet Krishna et al. (2017), DiDeMo Anne Hendricks et al. (2017), and LSMDC Rohrbach et al. (2015). Detailed descriptions of these datasets are provided in Appendix B.

Table 2: Comparisons in the text-to-video task on MSRVTT. We evaluate GFLOPs only for the video backbone during inference. A detailed comparison of memory usage can be found in Appendix C.7.

| Methods | | # Params (M) | GFLOPs | R@1↑ | R@5↑ | R@10↑ | R-sum↑ | MnR↓ |
|---|---|---|---|---|---|---|---|---|
| CLIP-ViT-B/32 | | | | | | | | |
| Full Fine-tuning | CLIP4Clip | 123.54 | 53.0 | 43.1 | 70.4 | 80.8 | 194.3 | 16.2 |
| Parameter-Efficient | Prompt | 0.08 | 58.2 | 40.4 | 66.3 | 77.3 | 184.0 | 16.7 |
| | Adapter | 0.26 | 53.1 | 41.9 | 69.9 | 78.7 | 190.2 | 14.9 |
| | LoRA | 0.49 | 53.0 | 43.7 | 68.9 | 80.4 | 193.0 | 16.0 |
| | PLEVU | 6.35 | - | 36.7 | 64.6 | 76.8 | 178.1 | - |
| | VoP$^{F+C}$ | 14.10 | 58.0 | 44.6 | 69.9 | 80.3 | 194.8 | 16.3 |
| | DGL$^L$ | 0.83 | 67.4 | 44.7 | 70.5 | 79.2 | 194.4 | 16.2 |
| | DGL$^T$ | 9.57 | >67.4 | 45.8 | 69.3 | 79.4 | 194.5 | 16.3 |
| Parameter-Efficient & Inference-Efficient | EVIT | 0.49 | 37.2 | 41.4 | 69.0 | 78.1 | 188.5 | 16.8 |
| | DiffRate | 0.49 | 36.8 | 41.5 | 68.6 | 79.8 | 189.9 | 16.3 |
| | STA | 0.49 | 35.7 | 42.6 | 69.5 | 78.8 | 190.9 | 17.0 |
| | ToMe | 0.49 | 40.2 | 42.9 | 68.3 | 80.2 | 191.4 | 16.2 |
| | TESTA | 0.59 | 40.6 | 43.7 | 69.0 | 79.4 | 192.1 | 16.8 |
| | TempMe | 0.50 | **34.8** | **46.1** | **71.8** | **80.7** | **198.6** | **14.8** |
| CLIP-ViT-B/16 | | | | | | | | |
| Parameter-Efficient | MV-Adapter | 3.6 | >210 | 46.0 | 72.0 | 82.1 | 200.1 | - |
| | RAP | 1.06 | >210 | 46.5 | 73.9 | 82.0 | 202.4 | 12.1 |
| | VoP | 14.10 | 246.2 | 47.7 | 72.4 | 82.2 | 202.3 | 12.0 |
| | DGL$^L$ | 0.83 | 251.2 | 48.3 | 71.8 | 80.6 | 200.7 | 13.4 |
| Param&Infer-Efficient | TempMe | 0.50 | **121.4** | **49.0** | **74.4** | **83.3** | **206.7** | **11.9** |

Table 3: Comparisons in the text-to-video task on ActivityNet, DiDeMo, and LSMDC with CLIP-VIT-B/32. CLIP4Clip is a classic full fine-tuning method.

| Methods | ActivityNet | | | | DiDeMo | | | | LSMDC | | | |
|---|---|---|---|---|---|---|---|---|---|---|---|---|
| | R@1↑ | R@5↑ | R@10↑ | MnR↓ | R@1↑ | R@5↑ | R@10↑ | MnR↓ | R@1↑ | R@5↑ | R@10↑ | MnR↓ |
| CLIP4Clip | 40.5 | 72.4 | - | 7.4 | 43.4 | 70.2 | 80.6 | 17.5 | 20.7 | 38.9 | 47.2 | 65.3 |
| Prompt | 36.0 | 67.0 | 79.4 | 10.2 | 39.4 | 67.6 | 78.3 | 20.6 | 19.1 | 38.2 | 46.1 | 67.5 |
| Adapter | 37.8 | 69.0 | 81.9 | 8.7 | 41.7 | 68.0 | 79.3 | 19.9 | 21.3 | 38.7 | 48.3 | 60.7 |
| LoRA | 38.7 | 70.6 | 82.2 | 8.7 | 41.6 | 68.8 | 78.6 | 20.8 | 20.7 | 39.5 | 47.4 | 68.3 |
| PLEVU | - | - | - | - | 36.1 | 64.8 | - | - | 13.4 | 29.5 | 40.3 | - |
| VoP$^{F+C}$ | 35.1 | 63.7 | 77.6 | 11.4 | 46.4 | 71.9 | 81.5 | **13.6** | 21.1 | 40.9 | 49.6 | 60.1 |
| DGL$^L$ | 38.6 | 69.2 | 81.6 | 9.0 | - | - | - | - | 21.4 | 39.4 | 48.4 | 64.3 |
| DiffRate | 38.0 | 69.1 | 81.6 | 9.2 | 40.8 | 67.8 | 77.8 | 20.6 | 20.2 | 37.4 | 47.6 | 70.0 |
| ToMe | 38.6 | 69.8 | 81.2 | 9.1 | 40.5 | 68.6 | 77.8 | 20.9 | 20.5 | 38.8 | 46.8 | 67.8 |
| TempMe | **44.9** | **75.2** | **85.5** | **6.8** | **48.0** | **72.4** | **81.8** | 13.7 | **23.5** | **41.7** | **51.8** | **53.5** |

**Metrics.** We evaluate the performance using metrics such as Recall at K (R@1, R@5, and R@10), the sum of these recalls (R-Sum), and Mean Rank (MnR). A higher recall score ↑ indicates better performance, while a lower MnR score ↓ denotes better performance. We evaluate GFLOPs (Giga Floating-Point Operations per Second) and throughput only for the video backbone during inference. All throughputs are measured on a A100.

**Implementation Details.** Following previous works Luo et al. (2022); Ju et al. (2022); Huang et al. (2023); Yang et al. (2024), we use the pre-trained CLIP as the backbone. We employ the AdamW optimizer Loshchilov & Hutter (2016) with a batch size of 128. The initial learning rate is set to 6e-4 with a cosine learning rate schedule Goyal et al. (2017) for 5 epochs. The dimension of LoRA is set to 8 in all experiments. The ImgMe Block employs $r = 2$ for CLIP-ViT-B/32 and $r = 10$ for CLIP-ViT-B/16. The ClipMe Block employs $R_C = 70\%, R_I = 90\%$ for ViT-B/32 and $R_C = 60\%, R_I = 80\%$ for CLIP-ViT-B/16. For short video retrieval datasets like MSRVTT and LSMDC, the max word and frame lengths are set to 32 and 12. 12 frames are merged into 6 clips at layer 9. These 6 clips are then merged into 3 clips at layer 10, and finally into a single clip at layer 11. The cross-clip merging step is skipped in the last layer. This process is denoted as $12 \xrightarrow{9} 6 \xrightarrow{10} 3 \xrightarrow{11} 1$. For long video retrieval datasets like ActivityNet and DiDeMo, the max word and frame lengths are set to 64 and 64. We employ $64 \xrightarrow{9} 16 \xrightarrow{10} 4 \xrightarrow{11} 1$. Unless noted otherwise, CLIP-ViT-B/32 is employed as the backbone.

**Compared Baselines.** We compare our TempMe with several strong baselines: (1) **CLIP4Clip** Luo et al. (2022), a classic full fine-tuning method. For a fair comparison, we only consider the parameter-free type that does not introduce additional modules. (2) **Prompt** Khattak

Table 4: Computational overhead comparisons of the CLIP-ViT-B/16 backbone. We evaluate the text-to-video accuracy metrics on MSRVTT.

| Methods | videos/s | GFLOPs | # Tokens | R@1/R-Sum↑ |
|---|---|---|---|---|
| Prompt | 29.7 | 216.8 | 2369 | 44.3/194.2 |
| Adapter | 29.4 | 211.7 | 2364 | 44.9/196.8 |
| LoRA | 30.6 | 211.3 | 2364 | 47.3/201.4 |
| VoP$^{F+C}$ | 25.0 | 246.2 | 2368 | 47.7/202.3 |
| DGL$^L$ | 3.3 | 251.2 | 2416 | 48.3/200.7 |
| DiffRate | 40.6 | 138.5 | 588 | 47.3/202.4 |
| ToMe | 40.8 | 144.4 | 924 | 46.2/200.6 |
| TempMe | **45.1** | **121.4** | **127** | **49.0/206.7** |

Table 5: Generalization analysis of full fine-tuning on MSRVTT with CLIP-ViT-B/16. The accuracy metrics are evaluated in the text-to-video task. Each experiment was conducted five times with different random seeds. $^\dagger$ denotes our own re-implementation.

| Methods | GFLOPs | Train Speed | Train Memory | Infer Memory | R@1/R-Sum↑ |
|---|---|---|---|---|---|
| CLIP4Clip$^\dagger$ | 211.3 | 1.00× | 70.1GB | 8.69GB | 47.3/202.3 |
| +ToMe | 144.4 | 1.35× | 61.6GB | **8.55GB** | 47.2/201.6 |
| +TempMe | **121.4** | **1.57×** | **52.7GB** | **8.55GB** | **50.9/210.2** |

Table 6: Application in video foundation models for text-video retrieval. We evaluate full fine-tuning on MSRVTT for the text-to-video task with UMT-B/16-25M.

| Methods | GFLOPs | Train Time | R@1↑ | R-Sum↑ |
|---|---|---|---|---|
| UMT | 304.3 | 9.5h | 51.0 | **212.3** |
| UMT4Clip | 210.1 | 4.8h | 48.8 | 206.7 |
| UMT4Clip+TempMe | **111.5** | **3.5h** | **51.1** | 209.2 |

Table 7: Application in video foundation models for text-video QA. We evaluate full fine-tuning on MSR-QA with UMT-B/16-25M.

| Methods | GFLOPs | Train Time | Accuracy |
|---|---|---|---|
| UMT | 304.3 | 6.3h | **44.9** |
| UMT4Clip | 210.1 | 3h | 44.4 |
| UMT4Clip+TempMe | **111.5** | **2.3h** | 44.6 |

et al. (2023), **LoRA** Hu et al. (2021), and **Adapter** Houlsby et al. (2019), which are mainstream parameter-efficient fine-tuning methods. (3) **PLEVU** Ju et al. (2022), **VoP** Huang et al. (2023), **DGL** Yang et al. (2024), **MV-Adapter** Jin et al. (2024), and **RAP** Cao et al. (2024), which are methods tailored for parameter-efficient text-video retrieval. (4) **EVIT** Liang et al. (2022), **DiffRate** Chen et al. (2023a), and **ToMe** Bolya et al. (2022), which are token compression methods for image models. (5) **STA** Ding et al. (2023) and **TESTA** Ren et al. (2023), which are video token compression methods. For fairness, token compression methods apply LoRA like our TempMe.

## 4.2 COMPARISONS WITH STATE-OF-THE-ART METHODS

**Efficient Fine-tuning.** In Table 2, we compare the t2v performance on MSRVTT. Our TempMe achieves significant improvements over previous methods, with a **3.8**% R-Sum increase using ViT-B/32 and a **4.3**% R-Sum increase using ViT-B/16, while maintaining minimal GFLOPs. Table 3 shows the t2v results on ActivityNet, DiDeMo, and LSMDC, demonstrating that TempMe consistently outperforms state-of-the-art methods. Detailed v2t results are provided in Appendix C.1.

**Computational Overhead.** In Table 4, accuracy metrics are evaluated on MSRVTT with CLIP-ViT-B/16. We also report the model throughput and complexity during the inference phase. During this phase, the tunable weights can be merged in LoRA, DiffRate, ToMe, and TempMe (see Section 3.1). For fairness, prompt generation in VoP and DGL is omitted when evaluating throughput and complexity. Compared to LoRA, ToMe reduces tokens per frame to **39**%, with a minor performance degradation. However, our TempMe further reduces temporal tokens across frames to just **5**%, while also improving the R-Sum by **5.3**%. Unlike VoP and DGL, which compromise efficiency for performance, our TempMe achieves a **1.8**× speedup over VoP and a **13.7**× speedup over DGL, while reducing GFLOPs by **51**% and improving R-Sum by **4.4**%.

## 4.3 GENERALIZATION ANALYSIS

**Full Fine-tuning.** Table 5 shows that our TempMe can be applied to full fine-tuning. The image encoder must process each sampled frame individually, resulting in significant GPU memory usage and training times. ToMe reduces spatial redundancy to boost model speed but suffers from a minor performance decrease. In contrast, our TempMe not only reduces temporal redundancy but also learns video-level features, with a **1.57**× speedup and a **7.9**% R-Sum gain.

**Application in Video Foundation Models.** We extend our method to video foundation methods, such as UMT Li et al. (2023). TempMe is built upon the text-image CLIP model, which processes each sampled frame individually. This contrasts with UMT, which processes all frame tokens simul-

Table 8: Ablation on each component in t2v on MSRVTT with CLIP-ViT-B/32. CLIP-straight denotes the zero-shot performance.

| Methods | GFLOPs | R@1↑ | R-Sum↑ | MnR↓ |
|---|---|---|---|---|
| CLIP-Straight | 53.0 | 30.8 | 147.9 | 41.8 |
| +LoRA | 53.0 | 43.7 | 193.0 | 16.0 |
| +ImgMe | 40.2 | 42.9 | 191.4 | 16.2 |
| +ClipMe | **34.8** | **46.1** | **198.6** | **14.8** |

Table 9: Ablation on each function in t2v on MSRVTT with CLIP-ViT-B/32. Temporal Modeling is shown in Figure 6 of Appendix.

| Methods | GFLOPs | R@1↑ | R-Sum↑ | MnR↓ |
|---|---|---|---|---|
| LoRA | 53.0 | 43.7 | 193.0 | 16.0 |
| Temporal Modeling | 54.3 | **46.3** | **199.7** | **14.8** |
| Token Reduction | **34.7** | 41.6 | 188.8 | 16.6 |
| TempMe | 34.8 | 46.1 | 198.6 | **14.8** |

taneously. To align with our method, we introduce a new baseline UMT4Clip, where UMT handles frames separately. These differences in token processing strategies inevitably lead to a decline in performance. Following the practice of UMT, we conduct fully fine-tuning experiments on both text-video retrieval and video QA tasks. In Table 6, our TempMe achieves performance close to UMT while requiring only **one-third** of the GFLOPs and training time. Moreover, we also extend our method to video QA in Table 7. These results demonstrate the generalizability of our method.

**More Generalization Analysis.** To further validate the generalization capability, we integrate our TempMe with other parameter-efficient fine-tuning methods such as Prompt and Adapter in Appendix C.2. Moreover, we evaluate our TempMe on other backbones. These detailed experimental results are provided in Appendix C.3.

## 4.4 ABLATION STUDY

We conduct an analysis of TempMe from structural and functional perspectives, respectively.

**Ablation on Each Component.** TempMe is architecturally divided into the ImgMe and ClipMe blocks. Table 8 illustrate the impact of each component. CLIP-straight denotes the zero-shot performance of CLIP, and LoRA is employed as the baseline for efficient fine-tuning. The ImgMe block reduces spatial redundancy in each frame. The ClipMe block aggregates short-frame clips into extended-frame clips, facilitating temporal learning while minimizing temporal redundancy. Finally, our TempMe outperforms LoRA by **5.6**% R-Sum and reduces **18.2** GFLOPs.

**Ablation on Each Function.** From a functional perspective, TempMe can be categorized into Temporal Modeling and Token Reduction, aimed at improving accuracy and efficiency, respectively. Table 9 demonstrates the impact of each function. (1) The accuracy improvements are attributed to Temporal Modeling, which aggregates clips progressively to enhance spatio-temporal learning. In this framework, the attention modules of the early layers are applied to intra-frame tokens in the spatial domain. In the later layers, they operate on tokens across frames in the spatio-temporal domain. (2) The efficiency improvements arise from Token Reduction, which reduces redundancy in the entire framework. In the early layers, it slightly reduces intra-frame tokens to decrease spatial redundancy. In the later layers, it significantly reduces tokens among frames to address large temporal redundancy. (3) Without the support of Temporal Modeling, Token Reduction alone significantly reduces tokens in the spatial domain, leading to a substantial **4.2**% decrease in R-sum. However, due to the considerable temporal redundancy, the combination of Temporal Modeling and Token Reduction (TempMe) significantly reduces complexity while achieving high performance.

**Ablation on Merging Strategy.** Table 10 validates the effectiveness of the merging strategy utilized in our PMG framework. (1) *Progressive Merging.* In A1, we observe that merging all clips at once leads to high model complexity, due to the quadratic complexity of MHSA with the surge in token numbers. Alternatively, the progressive merging of clips benefits video-level information. Our A0 with lower complexity achieves better performance. (2) *Holistic Merging.* We conduct experiments in A2 to evaluate the effectiveness of merging all clips. A2 shows that a notable performance decline when frames are not fully merged (e.g., 12 frames into 3 or 4 frames). These findings indicate that encoding all frame information simultaneously in MHSA is crucial for holistic video understanding. (3) *Continuous Merging.* Experiments on merging with various gaps are conducted in A3. Larger gaps trigger an earlier start of the clip merging process, which leads to a higher token reduction but also a drop in performance. In conclusion, we adopt progressive, holistic, and continuous merging (A0) as our final strategy.

Table 10: Ablation analysis of different merging strategies on MSRVTT with CLIP-VIT-B/32. We adopt progressive, holistic, and continuous merging (A0) as our final strategy.

| Methods | | GFLOPs | Text-to-Video | | | | Video-to-Text | | | |
|---|---|---|---|---|---|---|---|---|---|---|
| | | | R@1↑ | R@5↑ | R@10↑ | MnR↓ | R@1↑ | R@5↑ | R@10↑ | MnR↓ |
| A0 | $12 \xrightarrow{9} 6 \xrightarrow{10} 3 \xrightarrow{11} 1$ | 34.8 | **46.1** | **71.8** | 80.7 | **14.8** | **45.6** | **72.4** | 81.2 | **10.2** |
| A1 | $12 \xrightarrow{9} 4 \xrightarrow{10} 1$ | 35.4 | 45.7 | 71.2 | 80.5 | **14.8** | 45.0 | 70.8 | **81.7** | 11.1 |
| | $12 \xrightarrow{9} 1$ | 37.0 | 44.7 | 69.3 | 80.8 | 15.8 | 44.6 | 70.7 | 80.4 | 11.2 |
| A2 | $12 \xrightarrow{9} 6 \xrightarrow{10} 3$ | 35.4 | 43.6 | 70.2 | 79.8 | 15.9 | 43.5 | 72.1 | 81.0 | 11.4 |
| | $12 \xrightarrow{9} 4$ | 36.8 | 43.9 | 70.5 | **81.2** | 15.6 | 43.8 | 71.6 | 81.5 | 11.2 |
| A3 | $12 \xrightarrow{7} 6 \xrightarrow{9} 3 \xrightarrow{11} 1$ | 31.9 | 44.4 | 71.2 | 80.5 | 15.7 | 43.8 | 72.2 | **81.7** | 11.4 |
| | $12 \xrightarrow{4} 6 \xrightarrow{7} 3 \xrightarrow{10} 1$ | 25.8 | 42.5 | 69.5 | 79.2 | 15.5 | 42.6 | 70.3 | 80.4 | 11.3 |
| | $12 \xrightarrow{1} 6 \xrightarrow{5} 3 \xrightarrow{9} 1$ | 18.1 | 40.8 | 68.1 | 78.6 | 15.8 | 40.6 | 67.8 | 79.2 | 12.5 |

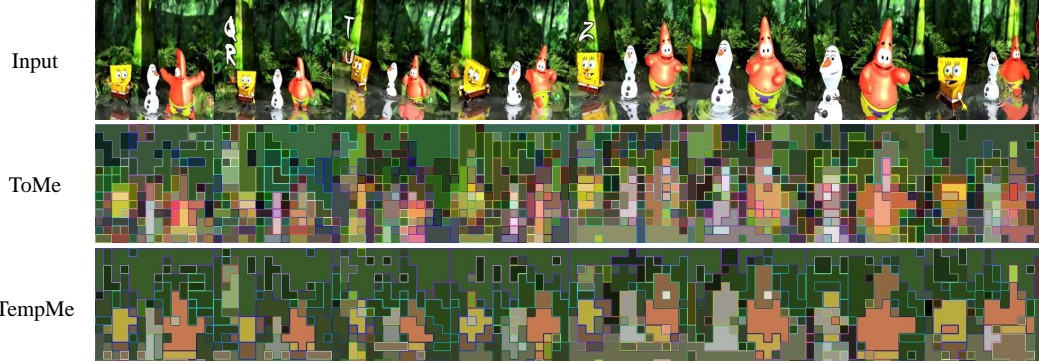

Figure 4: Qualitative comparisons on MSRVTT with CLIP-ViT-B/16. Patches that share the same inner and border color are merged. TempMe merges tokens of similar elements across frames.

**More Ablation Analysis.** We further conduct a series of ablation studies. First, since the trainable parameters of TempMe using CLIP-ViT-B/32 include LoRA (∼0.49M) and clip positional embeddings (∼0.01M), we perform ablation on the clip positional embeddings of the ClipMe block in Appendix C.4. Next, we provide a detailed analysis of the intra-clip and cross-clip merging steps within the ClipMe block in Appendix C.5. Furthermore, given that the number of sampled frames affects the model complexity, we conduct an ablation study on frame sampling in Appendix C.6. Finally, we report memory usage during training and inference in Appendix C.7.

## 4.5 QUALITATIVE RESULTS

Figure 4 presents visualization results of ToMe and our TempMe. ToMe only merges spatial tokens within individual frames, leaving a large number of redundant tokens across frames, contributing to high computational overhead. In contrast, TempMe merges tokens of similar elements across contiguous frames. For instance, TempMe merges similar body regions of Patrick Star across frames, such as the head, upper body, and lower body. These visualization results show that our TempMe effectively reduces temporal redundancy. More qualitative results are provided in Appendix C.8.

## 5 CONCLUSION

We explore temporal redundancy for efficient text-video retrieval. A parameter- and inference-efficient text-video retrieval method, Temporal Token Merging, is proposed to reduces model complexity while simultaneously extracting comprehensive video features. We introduce an innovative progressive multi-granularity framework to merge redundant tokens in consecutive clips and encode information from various frames. Extensive experiments validate the superiority and generalizability of TempMe. Further discussion on the limitations and broader impacts is provided in Appendix D. In the future, we aim to extend the application of our proposed method to other video tasks.

ACKNOWLEDGMENTS

This work was supported by the Key R & D Program of Xinjiang, China (2022B01006), Zhejiang Provincial Natural Science Foundation of China (No. LDT23F01013F01), the National Natural Science Foundation of China (No. 62441614), CCF-DiDi GAIA Collaborative Research Funds, China Postdoctoral Science Foundation (2024M750565), Guangdong S & T Program (2024B0101040008), and the Key Realm Research and Development Program of Guangzhou (No.2024B01W0007).

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

# A    PRELIMINARIES

**CLIP for Text-Video Retrieval.**    Following the approach of previous works Luo et al. (2022); Ju et al. (2022); Huang et al. (2023); Yang et al. (2024), we utilize the pre-trained text-image CLIP model Radford et al. (2021) as the backbone, which exhibits strong performance in downstream tasks. The full CLIP is composed of both a text encoder and an image encoder, employing the Transformer architecture Vaswani et al. (2017); Dosovitskiy et al. (2020) which consists of alternating blocks of multi-head self-attention (MHSA) and feed-forward network (FFN). The attention module in MHSA:

$$\text{Attention}(x) = \text{softmax}(\frac{(xQ)^T(xK)}{\sqrt{d}})(xV),  \tag{1}$$

where $Q \in \mathbb{R}^{D \times d}$, $K \in \mathbb{R}^{D \times d}$, and $V \in \mathbb{R}^{D \times d}$ are three projection matrices.

Given an input text $t$, the text encoder first tokenizes and transforms the text description into word tokens. Then, these tokens are processed through 12 text encoder layers to extract the final text feature $\mathbf{t}$. For an input video $v$, each sampled frame $I_i(i = 1, \cdots, F)$ is separately processed by the image encoder as shown in Figure 1b. The image $I$ is divided into fixed-size patches and projected into patch tokens, which are inputted into 12 image encoder layers to extract the frame feature $\mathbf{f}$. The final video feature $\mathbf{v}$ is obtained by averaging all frame features $\{\mathbf{f}_i\}_{i=1}^{F}$.

Finally, the cross-modal contrastive loss Oord et al. (2018) is applied to jointly optimize in both text-to-video and video-to-text directions:

$$\mathcal{L}_{t2v} = -\frac{1}{B} \sum_{i=1}^{B} \log \frac{\exp(s(t_i, v_i)/\tau)}{\sum_{j=1}^{B} \exp(s(t_i, v_j)/\tau)},  \tag{2}$$

$$\mathcal{L}_{v2t} = -\frac{1}{B} \sum_{i=1}^{B} \log \frac{\exp(s(t_i, v_i)/\tau)}{\sum_{j=1}^{B} \exp(s(t_j, v_i)/\tau)},  \tag{3}$$

$$\mathcal{L} = (\mathcal{L}_{t2v} + \mathcal{L}_{v2t})/2,  \tag{4}$$

where $B$ is the batch size, $\tau$ is the temperature hyper-parameters, and $s(t, v) = \frac{\mathbf{t}^T \mathbf{v}}{\|\mathbf{t}\|\|\mathbf{v}\|}$ is the cosine similarity. This loss function maximizes the similarity of corresponding text-video pairs.

**LoRA.**    LoRA Hu et al. (2021) is a parameter-efficient method to adapt CLIP with minimal training. LoRA injects trainable rank decomposition matrices into the attention module of each layer. For the pre-trained matrices $W \in \mathbb{R}^{D \times d}$ in $Q$, $K$, and $V$ of Eq. (1), LoRA optimize their rank-decomposed changes, $\Delta W = W^{down}W^{up}$, where $W^{down} \in \mathbb{R}^{D \times r}$, $W^{up} \in \mathbb{R}^{r \times d}$, and the rank $r \ll \min(D, d)$. For $\bar{x} = xW$, the forward pass is modified to $\bar{x} = xW + x\Delta W$. The trainable matrices $\Delta W$ can be merged with the frozen weights $W$, introducing no extra inference latency.

# B    ADDITIONAL EXPERIMENTAL SETTINGS

**Datasets.**    We evaluate on four benchmark datasets: MSR-VTT Xu et al. (2016), ActivityNet Krishna et al. (2017), DiDeMo Anne Hendricks et al. (2017), and LSMDC Rohrbach et al. (2015). (1) **MSR-VTT** Xu et al. (2016) is comprised of 10,000 YouTube videos, each paired with 20 text descriptions. Following the data split in Gabeur et al. (2020); Miech et al. (2019), we train models on 9000 train+val videos with the corresponding captions and test on the 1K-A test set with 1000 video-text pairs. (2) **ActivityNet** Krishna et al. (2017) contains 20,000 YouTube videos. We evaluate models on the 'val1' split, comprising 10,009 videos for training and 4,917 for testing. We follow the setting from Gabeur et al. (2020); Zhang et al. (2018), where all sentence descriptions for a video are concatenated into a single query. (3) **DiDeMo** Anne Hendricks et al. (2017) contains 10,000 videos annotated with 40,000 text descriptions. There are 8,395 videos in the train set and 1,004 videos in the test set. Following the setting in Bain et al. (2021); Lei et al. (2021), we concatenate all the descriptions of a video to form a paragraph and evaluate models for paragraph-video

Table 11: Comparisons in the video-to-text task on MSRVTT.

| Methods | | # Params (M) | GFLOPs | R@1↑ | R@5↑ | R@10↑ | R-sum↑ | MnR↓ |
|---|---|---|---|---|---|---|---|---|
| | | CLIP-ViT-B/32 | | | | | | |
| Full Fine-tuning | CLIP4Clip | 123.54 | 53.0 | 43.1 | 70.5 | 81.2 | 194.8 | 12.4 |
| Parameter-Efficient | Prompt | 0.08 | 58.2 | 42.2 | 69.7 | 79.2 | 191.1 | 12.4 |
| | Adapter | 0.26 | 53.1 | 43.6 | 69.9 | 80.1 | 193.6 | 11.5 |
| | LoRA | 0.49 | 53.0 | 43.0 | 70.2 | **82.2** | 195.4 | 12.0 |
| | DGL$^L$ | 0.83 | 67.4 | 42.1 | 70.0 | 80.6 | 192.7 | 13.4 |
| | VoP$^{F+C}$ | 14.10 | 58.0 | 44.5 | 70.7 | 80.6 | 195.8 | 11.5 |
| Parameter-Efficient & Inference-Efficient | EVIT | 0.49 | 37.2 | 43.5 | 69.4 | 80.2 | 193.1 | 11.7 |
| | DiffRate | 0.49 | 36.8 | 43.6 | 70.1 | 81.2 | 194.9 | 12.2 |
| | STA | 0.49 | 35.7 | 41.1 | 69.7 | 81.1 | 191.9 | 12.7 |
| | ToMe | 0.49 | 40.2 | 42.5 | 69.1 | 80.6 | 192.2 | 12.5 |
| | TESTA | 0.59 | 40.6 | 43.0 | 70.4 | 80.5 | 193.9 | 12.5 |
| | TempMe | 0.50 | **34.8** | **45.6** | **72.4** | 81.2 | **199.2** | **10.2** |
| | | CLIP-ViT-B/16 | | | | | | |
| Parameter-Efficient | MV-Adapter | 3.6 | >210 | 45.6 | 74.0 | 83.8 | 203.4 | - |
| | RAP | 1.06 | >210 | 45.3 | **76.4** | 84.8 | 206.5 | 9.1 |
| | DGL$^L$ | 0.83 | 251.2 | 45.7 | 74.0 | 82.9 | 202.6 | 10.9 |
| Param&Infer-Efficient | TempMe | 0.50 | **121.4** | **47.6** | 75.3 | **85.4** | **208.3** | **9.0** |

Table 12: Comparisons in the video-to-text task on ActivityNet, DiDeMo, and LSMDC with CLIP-ViT-B/32. CLIP4Clip is a classic full fine-tuning method.

| Methods | ActivityNet | | | | DiDeMo | | | | LSMDC | | | |
|---|---|---|---|---|---|---|---|---|---|---|---|---|
| | R@1↑ | R@5↑ | R@10↑ | MnR↓ | R@1↑ | R@5↑ | R@10↑ | MnR↓ | R@1↑ | R@5↑ | R@10↑ | MnR↓ |
| CLIP4Clip | 42.5 | 74.1 | 85.8 | 6.6 | 42.5 | 70.6 | 80.2 | 11.6 | 20.6 | 39.4 | 47.5 | 56.7 |
| Prompt | 38.4 | 68.8 | 81.1 | 9.3 | 39.6 | 68.2 | 78.5 | 12.4 | 19.9 | 37.6 | 46.6 | 58.2 |
| Adapter | 40.0 | 71.0 | 82.9 | 7.7 | 42.7 | 70.4 | 79.4 | 12.3 | 20.5 | 38.7 | 48.7 | 53.6 |
| LoRA | 40.0 | 71.4 | 82.9 | 7.9 | 41.2 | 69.0 | 79.2 | 12.5 | 20.4 | 37.3 | 47.1 | 59.7 |
| VoP$^{F+C}$ | 35.6 | 65.9 | 77.8 | 10.4 | 44.4 | 71.8 | 81.8 | 9.5 | **22.3** | 40.3 | 50.7 | 51.1 |
| DiffRate | 39.1 | 69.7 | 82.3 | 8.4 | 40.4 | 67.5 | 79.0 | 12.6 | 20.0 | 37.8 | 46.9 | 61.6 |
| ToMe | 39.7 | 70.2 | 82.4 | 8.3 | 40.1 | 68.5 | 78.9 | 12.9 | 20.2 | 37.4 | 46.3 | 60.2 |
| TempMe | **45.3** | **74.7** | **86.2** | **6.4** | **48.4** | **75.4** | **83.6** | **9.1** | 22.2 | **41.5** | **51.5** | **48.0** |

retrieval. (4) **LSMDC** Rohrbach et al. (2015) is a movie clip dataset containing 118081 videos each paired with a single text description. 101,079 videos are used for training. 7,408 and 1,000 videos are used for validation and testing, respectively. The results of the test set are reported.

# C  ADDITIONAL EXPERIMENTAL RESULTS

## C.1  COMPARISONS IN THE VIDEO-TO-TEXT TASK

In Table 11, we compare the v2t performance on MSRVTT. In Table 12, we compare the v2t performance on ActivityNet, DiDeMo, and LSMDC. Consistent with the t2v comparisons detailed in the man paper, our TempMe achieves state-of-the-art performance with minimal computational costs.

## C.2  GENERALIZATION ANALYSIS OF DIFFERENT PARAMETER-EFFICIENT METHODS

In our main paper, our TempMe employs LoRA for parameter-efficient adaption. Moreover, our TempMe is compatible with other parameter-efficient fine-tuning methods, such as Prompt and Adapter, as shown in Tabel 13. We apply the same hyper-parameters (see Section 4.1) for both Prompt and Adapter. The extra ∼0.01 tunable parameters of TempMe are introduced by the tunable clip positional embeddings (see Figure 3). These results indicate that our TempMe can effectively integrate with various parameter-efficient methods.

## C.3  GENERALIZATION ANALYSIS OF DIFFERENT BACKBONES

To further assess the generalization capability of TempMe, we employed additional text-image pre-trained models as the backbone. In Table 14, we utilize ViT-B/16 as the backbone architecture,

Table 13: Generalization analysis of different parameter-efficient fine-tuning methods on MSRVTT with CLIP-ViT-32.

| Methods | # Params (M) | Text-to-Video | | | | Video-to-Text | | | |
|---|---|---|---|---|---|---|---|---|---|
| | | R@1↑ | R@5↑ | R@10↑ | MnR↓ | R@1↑ | R@5↑ | R@10↑ | MnR↓ |
| Prompt | 0.08 | 40.4 | 66.3 | 77.3 | 16.7 | 42.2 | 69.7 | 79.2 | 12.4 |
| +ToMe | 0.08 | 39.9 | 67.2 | 78.0 | 17.0 | 41.0 | 69.7 | 78.7 | 13.4 |
| +TempMe | 0.09 | **44.4** | **70.0** | **80.5** | **15.1** | **44.7** | **72.7** | **82.0** | **11.3** |
| Adapter | 0.26 | 41.9 | 69.9 | 78.7 | 14.9 | 43.6 | 69.9 | 80.1 | 11.5 |
| +ToMe | 0.26 | 41.3 | 68.7 | 79.7 | 15.1 | 42.4 | 71.1 | 79.5 | 12.0 |
| +TempMe | 0.28 | **45.7** | **71.2** | **80.1** | **14.1** | **44.6** | **72.1** | **81.1** | **11.2** |
| LoRA | 0.49 | 43.7 | 68.9 | 80.4 | 16.0 | 43.0 | 70.2 | **82.2** | 12.0 |
| +ToMe | 0.49 | 42.9 | 68.3 | 80.2 | 16.2 | 42.5 | 69.1 | 80.6 | 12.5 |
| +TempMe | 0.50 | **46.1** | **71.8** | **80.7** | **14.8** | **45.6** | **72.4** | 81.2 | **10.2** |

Table 14: Generalization analysis of different backbones on MSRVTT.

| Backbone | Methods | GFLOPs | Text-to-Video | | | | Video-to-Text | | | |
|---|---|---|---|---|---|---|---|---|---|
| | | | R@1↑ | R@5↑ | R@10↑ | MnR↓ | R@1↑ | R@5↑ | R@10↑ | MnR↓ |
| LaCLIP | LoRA | 211.3 | 44.5 | **71.4** | 79.7 | 14.4 | 43.9 | 71.2 | **82.9** | **11.1** |
| | ToMe | 144.4 | 43.4 | 70.9 | 80.7 | 16.0 | 43.9 | 71.2 | 82.8 | 11.5 |
| | TempMe | **121.4** | **47.2** | 70.8 | **81.4** | **14.2** | **46.5** | **73.1** | 82.7 | **11.1** |
| MetaCLIP | LoRA | 211.3 | 45.3 | 71.2 | 80.6 | 13.8 | 46.5 | 73.7 | 82.2 | 11.5 |
| | ToMe | 144.4 | 44.1 | 70.8 | 80.6 | 14.8 | 44.6 | 73.5 | 82.3 | 11.7 |
| | TempMe | **121.4** | **48.2** | **72.6** | **82.0** | **13.0** | **47.2** | **73.9** | **83.0** | **10.0** |

Table 15: Ablation on clip positional embeddings in t2v on MSRVTT with CLIP-ViT-B/32. CPE indicates **C**lip **P**ositional **E**mbeddings. The parameters of trainable CPE are approximately 0.01M.

| Methods | #Params (M) | R@1↑ | R-Sum↑ | MnR↓ |
|---|---|---|---|---|
| TempMe | 0.50 | **46.1** | **198.6** | **14.8** |
| $w/o$ CPE | 0.49 | 45.7 | 197.6 | 15.5 |

Table 16: Ablation on frame sampling in t2v on MSRVTT with CLIP-ViT-B/32. $F = N$ indicates the sampling of $N$ frames from a video.

| Methods | GFLOPs | R@1↑ | R-Sum↑ | MnR↓ |
|---|---|---|---|---|
| LoRA $F = 8$ | 35.4 | 42.1 | 191.6 | 16.0 |
| TempMe $F = 8$ | **25.0** | **44.4** | **195.6** | **15.6** |
| LoRA $F = 12$ | 53.0 | 43.7 | 193.0 | 16.0 |
| TempMe $F = 12$ | **34.8** | **46.1** | **198.6** | **14.8** |

maintaining the same hyper-parameters for both LaCLIP Fan et al. (2024) and MetaCLIP Xu et al. (2024a) as used in CLIP. Due to utilizing the same Transformer architecture as CLIP, the model complexity is consistent with that of CLIP. These results indicate that our TempMe can effectively integrate with various text-image pre-trained backbones.

## C.4 Ablation on Clip Positional Embeddings

The trainable parameters of TempMe using CLIP-ViT-B/32 include LoRA (∼0.49M) and clip positional embeddings (∼0.01M). Clip positional embeddings are only added at the cross-clip merging step. Specifically, clip positional embeddings are added just before the tokens are merged. When tokens from different clips are merged, their respective clip positional embeddings are also merged, effectively preserving the temporal information. Table 15 presents an ablation study of clip positional embeddings, which improves 0.4% R1 and 1.0% R-Sum.

## C.5 Hyper-parameters Analysis of ClipMe Block

We conduct controlled experiments to identify the effect of hyper-parameters in our proposed ClipMe Block. (1) *Start Layer of Clip Merging.* Figure 5a shows the impact of $\text{Start}_{\text{clip}}$. Merging at earlier layers leads to noticeable performance degradation. This is likely because early layers in the CLIP image encoder capture low-level features, which are critical for frame-specific details but inadequate for modeling temporal relationships. In contrast, merging in later layers takes advantage of more semantic features that are better suited for learning spatio-temporal relationships. Larger

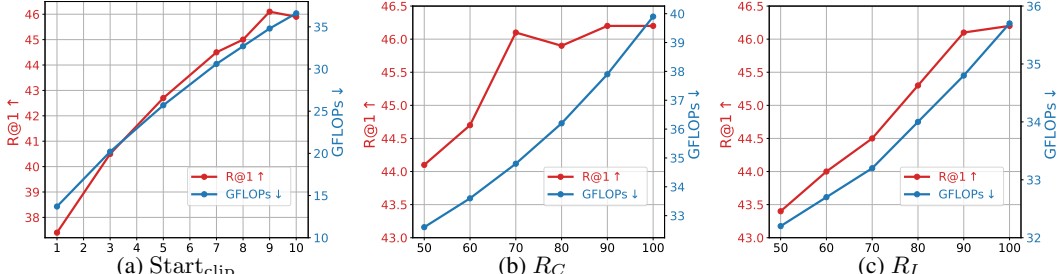

Figure 5: Hyper-parameters analysis for text-to-video results on MSR-VTT with VIT-B/32. Each hyper-parameter is evaluated while keeping all other hyper-parameters fixed.

Table 17: Memory usage comparisons in the text-to-video task on MSRVTT with CLIP-ViT-B/32.

| Methods | GFLOPs | Train Memory (MB) | Infer Memory (MB) | R@1 | R-Sum |
|---|---|---|---|---|---|
| Prompt | 58.2 | 16103 | 4941 | 40.3 | 184.0 |
| Adapter | 53.1 | 15291 | 4785 | 41.9 | 190.2 |
| LoRA | 53.0 | 16047 | 4783 | 43.7 | 193.0 |
| $\text{VoP}^{F+C}$ | 58.0 | 18913 | 6275 | 44.6 | 194.8 |
| $\text{DGL}^{L}$ | 67.4 | 18863 | 5203 | 44.7 | 194.4 |
| ToMe | 40.2 | 13701 | 4737 | 42.9 | 191.4 |
| TempMe | **34.8** | **12381** | **4735** | 46.1 | 198.6 |
| TempMe (F=18) | 51.5 | 16410 | 5879 | **47.9** | **200.9** |

$\text{Start}_{\text{clip}}$ enhances performance until it reaches 9, which is the most efficient value for maintaining high performance with lower complexity. (2) *Proportion of Remain Tokens.* $R_C$ and $R_I$ denote the proportions of remaining tokens for cross-clip and intra-clip merging, respectively. $R_* = 50\%$ means that half of the tokens are merged. $R_* = 100\%$ implies that the merging process is skipped. We seek a smaller $R_*$ that does not compromise accuracy. Figure 5b and 5c reveal that the optimal trade-off between performance and complexity is achieved at $R_C = 70\%, R_I = 90\%$.

### C.6 ABLATION ON FRAME SAMPLING

Following the standard protocol of previous works, we sample 12 frames per video on MSRVTT. Reducing the number of sampled frames would decrease computational complexity. We conduct experiments with 8 frames per video in Table 16. Although the reduction from 12 to 8 frames decreased the GFLOPs by 33%, it also caused a 1.4% decrease in R-sum. Despite this, our TempME remains effective even when only 8 frames are sampled, indicating its robustness.

### C.7 ABLATION ON MEMORY USAGE

In Table 17, we present the memory usage during training and inference for CLIP-ViT-32. For all experiments, memory usage during training and inference is measured on 4 A100 GPUs, each processing a batch size of 32.

(1) Although our TempMe achieves lower GFLOPs, it exhibits comparable inference memory usage to ToMe. We speculate that this is partially influenced by the quadratic complexity of attention mechanisms, particularly the maximum token length processed (denoted as Attn Capacity). Specifically, ToMe's Attn Capacity is 50 at layer 1, while our TempMe's Attn Capacity is 118 at layer 11. Additionally, a similar trend in memory usage is observed in full fine-tuning with CLIP-ViT-16 in Table 5. Despite comparable inference memory usage, our TempMe achieves a significant performance improvement, with a 3.2% increase in R@1 and a 7.2% increase in R-Sum, demonstrating its overall effectiveness.

(2) Compared to previous PEFT TVR methods such as VoP and DGL, our TempMe requires significantly less memory during training and inference, validating the efficiency of our memory optimiza-

tion. Specifically, Our TempMe not only improves 1.4% R@1 but also achieves greater efficiency in terms of both model complexity and memory usage.

(3) Increasing the input frames to 18 significantly improves performance by leveraging more temporal information. However, this comes at the cost of higher GFLOPs and memory usage, as each GPU needs to process 32x18 frames. Compared to previous PEFT TVR methods like VoP and DGL, our TempMe with 18 frames achieves a significant improvement of 3.2% R@1 and 6.1% R-sum under comparable inference memory usage. These results highlight the effectiveness of our TempMe in utilizing memory resources efficiently while achieving superior performance.

### C.8    ADDITIONAL VISUALIZATION RESULTS

In Figure 7, we present more visualization results. Regions sharing the same color in different frames are merged together. Through the boundaries of the merged regions, we can vaguely identify the objects they depict. The merged regions in TempMe are larger than those in ToMe. Moreover, although TempMe merges a greater number of tokens than ToMe, TempMe consistently outperforms ToMe in accuracy, confirming its advantage over ToMe.

## D    ADDITIONAL DISCUSSIONS

For efficient text-video retrieval, we propose TempMe, a novel method to reduce temporal redundancy when utilizing the pre-trained image encoder for video feature extraction. Our research employs publicly available datasets, which are restricted to non-commercial applications.

### D.1    LIMITATIONS

At present, we concentrate on accelerating video processing. However, there is potential for further improving the text encoder. Therefore, we suggest that an efficient method for both text and image encoders could be a valuable direction for further research.

When adapting our method to video foundation models (see Section 4.3 of our main paper), the transfer is only feasible when the video encoder can be treated as an image encoder. Specifically, our TempMe is successfully extended to UMT Li et al. (2023). However, it faces challenges when applied to specialized video architectures, particularly CLIP-ViP Xue et al. (2022) which uses video proxies to process all tokens simultaneously, making it less compatible with our TempMe.

### D.2    BROADER IMPACTS

The efficient text-video retrieval task focuses on efficiently transferring foundation models like CLIP. Previous studies Huang et al. (2023); Yang et al. (2024); Jin et al. (2024); Cao et al. (2024) focus on freezing the backbone and training only minimal tunable parameters. However, these studies often neglect the importance of model complexity and inference efficiency. Our research encourages the community to consider both model storage and speed during transfer.

Our proposed method increases the efficiency of retrieving relevant video content based on textual queries, leading to substantial savings in time and resources. In an era where video content dominates digital media, efficiently locating relevant videos is invaluable. Nonetheless, it also raises concerns about potential misuse in surveillance and monitoring, as it simplifies the extraction and analysis of video content from various sources.

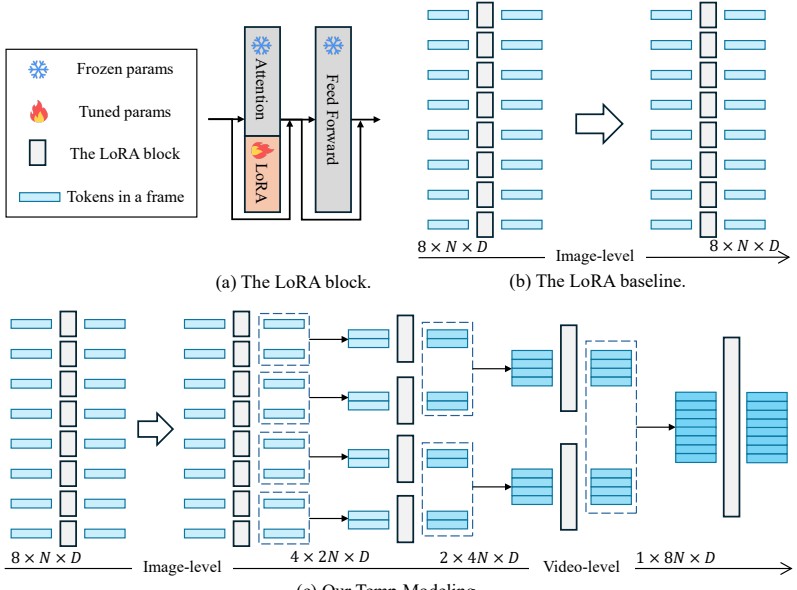

(a) The LoRA block.

(b) The LoRA baseline.

(c) Our Temp Modeling.

Figure 6: A toy example with 8 sampled frames as input. (a) LoRA injects trainable weights into the attention module of each layer. (b) LoRA processes each frame individually. (c) Our Temporal Modeling aggregates clips progressively to enhance spatio-temporal learning. In this framework, the self-attention modules of the early layers is applied to intra-frame tokens in the spatial domain, targeting image-level details. In the later layers, they operate on tokens across frames in the spatio-temporal domain, capturing video-level information. The PMG framework, shown in Figure 2 of the main paper, has Temporal Modeling as a core function.

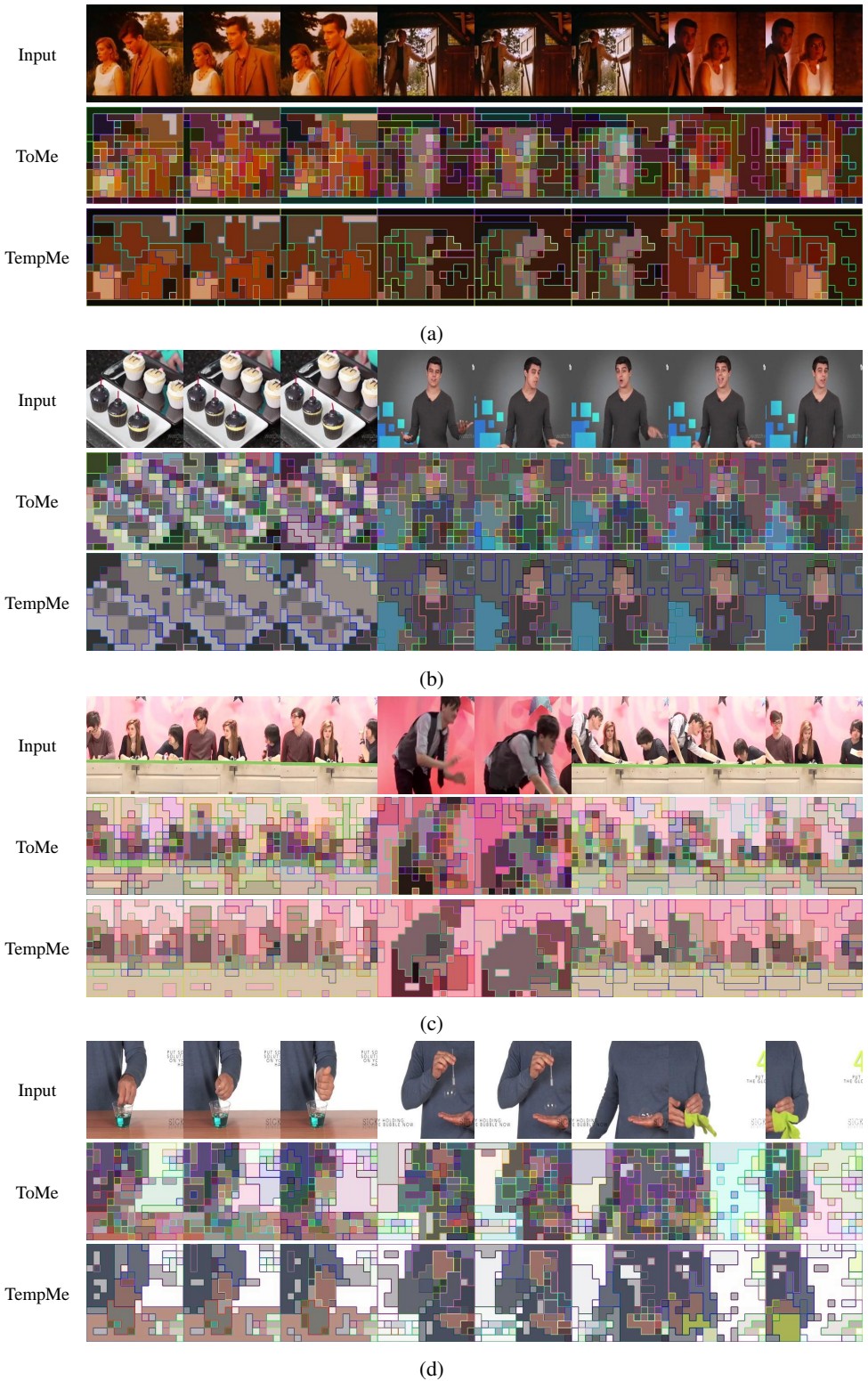

Figure 7: Qualitative comparisons on MSRVTT with CLIP-ViT-B/16. Patches that share the same inner and border color are merged. TempMe merges tokens of similar elements across frames.

