# OpenReview forum: "TempMe: Video Temporal Token Merging for Efficient Text-Video Retrieval"
_ICLR.cc/2025/Conference — ICLR 2025 Poster_

### Official Review · Reviewer_JLDW · 2024-11-01

**Soundness:** 3
**Presentation:** 2
**Contribution:** 3
**Rating:** 6
**Confidence:** 3

**Summary:**

This paper tackles the text-video retrieval task and specifically propose a parameter efficient method for this task. The proposed framework merges redundant tokens across adjacent video clips. The method has a good trade-off between inference and parameter efficiency and accuracy achieving state of the art results when compared to other parameter efficient methods.

**Strengths:**

The paper addresses an important task, namely text-video retrieval and proposes a parameter-efficient method for this. One strength of the paper is that the proposed method was extended to video foundation methods such as UMT and various backbones showcasing the extensibility of the method. Also, the method is based on a well-known fact, namely the video has a lot of redundant information from frame to frame and the method is built upon that observation and compresses the redundant information. Additionally the method achieves good results both in term of performance and inference efficiency.

**Weaknesses:**

- Abstract: the abstract is hard to follow and is not clear if the method addressed inference speed-ups or training time speed-ups or both.
- overall the quality of the writing can be improved because it's not straightforward to follow the paper, for example in the introduction the transitions between paragraphs are abrupt.
- the choice of sampling 12 frames for MSRVTT and 64 frames for ActivityNet seems arbitrary.

**Questions:**

Is any part of the method that is specifically designed for text-video retrieval or could it be applied for other text-video tasks as well?
[1] shares a similar idea, can you please summarize the similarities and differences between the proposed method and [1]?


[1] Yao, Linli, et al. "DeCo: Decoupling Token Compression from Semantic Abstraction in Multimodal Large Language Models." arXiv preprint arXiv:2405.20985 (2024).

---

> ### Author Response · Authors · 2024-11-22
> **Author Response to Reviewer JLDW**
>
> Thank you very much for positive and thoughtful comments,  which have been instrumental in refining and enhancing the quality and clarity of our manuscript.
>
> **Question 1**: Abstract: the abstract is hard to follow and is not clear if the method addressed inference speed-ups or training time speed-ups or both.
>
> **Answer**: We appreciate your comment on the clarity of the abstract. In the revised abstract, we have refined it for improved clarity and coherence. Specifically, we now emphasize that our method achieves efficiency in both training and inference while maintaining parameter efficiency. We have replaced "inference-efficient" with "training-inference efficient" to precisely convey this dual focus. This updated terminology will also be consistently applied throughout the paper in future revisions to enhance overall clarity and coherence.
>
> **Question 2**: Overall the quality of the writing can be improved because it's not straightforward to follow the paper, for example in the introduction the transitions between paragraphs are abrupt.
>
> **Answer**: Thank you for pointing out this issue. In the Introduction section of our revised paper, we have improved the transitions between paragraphs in the Introduction section to ensure smoother flow and better readability. Additionally, we will conduct a comprehensive review of the entire manuscript, using writing tools and feedback from native speakers, to further enhance clarity and readability in the final version.
>
> **Question 3**: The choice of sampling 12 frames for MSRVTT and 64 frames for ActivityNet seems arbitrary.
>
> **Answer**: Thank you for your observation. The number of sampled frames (12 for MSRVTT and 64 for ActivityNet) is chosen to align with previous methods like DGL, as mentioned in Lines 303–305 of our manuscript. This choice facilitates a fair comparison of our method with previous methods.
>
> **Question 4**: Is any part of the method that is specifically designed for text-video retrieval or could it be applied for other text-video tasks as well?
>
> **Answer**: Thank you for your question. While the primary focus of our method is text-video retrieval (TVR), its design is inherently versatile and can be employed for other text-video tasks. Specifically, we focus on a frame-wise processing paradigm in TVR, where each sampled frame is processed by pretrained CLIP. Our proposed TempMe progressively merges different frames, achieveing both efficiency and performance improvements. This design can be applied for other text-video tasks that utilize a similar frame-wise processing paradigm. As shown in Table 6 and 7 of our manuscript, we successfully apply TempMe to VideoQA task and integrate it with video backbones like UMT, demonstrating its generalizability and broader applicability.
>
> **Question 5**: [1] shares a similar idea, can you please summarize the similarities and differences between the proposed method and [1]? [1]Yao, Linli, et al. "DeCo: Decoupling Token Compression from Semantic Abstraction in Multimodal Large Language Models." arXiv preprint arXiv:2405.20985 (2024).
>
> **Answer**: Thanks for the supplement. In our revised paper, we have discussed DeCo in the Related Work section. DeCo adopts 2D Adaptive Pooling to downsample vision tokens at the spatial level for MLLMs. In contrast, our TempMe concentrates on temporal-level processing for text-video retrieval, which effectively learns temporal information and efficiently reduces temporal redundancy.

---

> ### Author Response · Authors · 2024-11-25
> **Official Comment by Authors**
>
> We sincerely appreciate your time and effort in reviewing our paper. We would like to kindly inquire if our revisions and responses have addressed your concerns. Your further feedback would be invaluable.

---

> > ### Comment · Reviewer_JLDW · 2024-11-25
> >
> > The rebuttal addressed my concerns, thus I will keep my positive rating and I incline towards acceptance for this paper.

---

> ### Author Response · Authors · 2024-11-25
> **Official Comment by Authors**
>
> We sincerely appreciate your positive feedback and are delighted that our responses have addressed your concerns. Your inclination towards acceptance is encouraging, and we value your support for our work.

---

### Official Review · Reviewer_A4xM · 2024-11-02

**Soundness:** 3
**Presentation:** 2
**Contribution:** 2
**Rating:** 6
**Confidence:** 5

**Summary:**

This work proposes TempMe, a temporal token merging method for T2V retrieval, designed to address two main issues: (1) high inference costs in current PEFT methods, and (2) the tendency of token merging methods to overlook temporal redundancy. TempMe introduces a PMG framework, which first merges tokens within individual images and then progressively merges across video clips. Experimental results demonstrate TEMP’s effectiveness, showing improvements in GFLOPS, training time, and retrieval performance.

**Strengths:**

1. **Efficient Training/Inference Acceleration**: TempMe achieves significant improvements in training efficiency by implementing image merging followed by progressive video clip merging. This approach leads to substantial reductions in GFLOPS and training time compared to previous methods.

2. **Comprehensive Ablation Studies**: Extensive ablation experiments validate the effectiveness of clip merging at different layers and intervals, as well as its applicability on stronger video-pretrained backbones like UMT and MetaCLIP. Qualitative results further show that TEMP effectively merges objects and backgrounds.

3. **Cross-Clip Merging**: TempMe introduces a relatively novel approach by incorporating cross-clip (inter-clip) merging, expanding beyond traditional intra-clip merging to capture cross-clip information, which enhances the learning of temporal relations across clips.

**Weaknesses:**

1. **Limited Improvement in R@1 for T2V Retrieval**: The R@1 improvement over previous methods is relatively small, especially given the advances in MLLM models. In 2023, T2V retrieval methods like HBI and Cap4video on CLIP-ViT-32/16, R@1 have reached around 48/50 (e.g., [1][2]). I suspect the limited improvement may stem from TempMe still focusing on video merging within the encoder, without further optimization after obtaining the clip-level representation.

2. **Lack of Memory Usage Comparison**: Efficiency is not solely about GFLOPS—memory usage for training and inference is also crucial. While TempMe provides training acceleration, memory optimization would be even more beneficial. In Table 5, the authors report memory usage for CLIP-ViT-16, but it would be helpful to see detailed comparisons for CLIP-ViT-32 as well, particularly relative to previous methods.

3. **Limited Novelty**: The intra-clip merging process is highly similar to the original token merging approach [3], which limits the novelty of this method. In fact, the flowchart for intra-clip merging in Figure 3 closely resembles the diagram in Token Merging (ToMe), making it difficult to discern substantial differences. Essentially, the only distinct contribution is the cross-clip merging mechanism.

4. **Efficiency and Early Layer Merging**: Performing token merging in earlier layers of CLIP could theoretically reduce both GFLOPS and memory usage, though it might harm performance metrics like R@1. TempMe, however, only applies merging in the last three layers with a 12-6-3-1 merging pattern. A potential improvement could be to increase the input frames and match GFLOPS with previous PEFT methods, potentially enhancing R@1 without excessive memory usage. Additionally, it would be beneficial to provide memory usage details for CLIP-ViT-32 to understand whether further optimization is feasible.

5. **Metric Suggestion**: Presenting an overall sum@R can be unclear for readers. R@1 is generally a more meaningful metric than sum@R and would offer a clearer understanding of the model's performance.

[1] Jin, P. et al., "Video-Text As Game Players: Hierarchical Banzhaf Interaction for Cross-Modal Representation Learning," CVPR, 2023.
[2] Wu, W. et al., "Cap4Video: What Can Auxiliary Captions Do for Text-Video Retrieval?" CVPR, 2023.
[3] Bolya, D. et al., "Token Merging: Your ViT but Faster," ICLR, 2023.

**Questions:**

If the memory usage of TempMe proves to be more efficient compared to previous PEFT or full finetuning methods, I would consider increasing my score.

**Details Of Ethics Concerns:**

no ethics concerns

---

> ### Author Response · Authors · 2024-11-22
> **Author Response to Reviewer A4xM (1/3)**
>
> We sincerely appreciate your valuable and constructive feedback, which has greatly assisted us in enhancing the overall quality of our manuscript.
>
> **Question 1.1**: Limited Improvement in R@1 for T2V Retrieval. The R@1 improvement over previous methods is relatively small, especially given the advances in MLLM models. In 2023, T2V retrieval methods like HBI and Cap4video on CLIP-ViT-32/16, R@1 have reached around 48/50 (e.g., Cap4Video, HBI).
>
> **Answer**: Thank you for your observation. It is important to note that methods like HBI and Cap4video perform **full fine-tuning**, where the entire backbone is trained (as discussed in the Related Work section of our revised paper). In this paper, we focus on **parameter-efficient fine-tuning** (PEFT) methods, where the backbone remains frozen, and only minimal trainable parameters are used. Compared to previous SOTA methods, our TempMe achieve both superior performance and computational efficiency. Specifically, compared to VoP, we achieve a **1.4%** R@1 improvement with only **3.6%** trainable parameters and **60%** GFLOPs. Similarly, compared to DGL, we achieve a **1.5%** R@1 improvement with only **60%** trainable parameters and **52%** GFLOPs.
>
> **Question 1.2**: I suspect the limited improvement may stem from TempMe still focusing on video merging within the encoder, without further optimization after obtaining the clip-level representation.
>
> **Answer**: Thank you for your insightful observation. For fair comparisons, our TempMe is consistent with existing PEFT TVR methods in computing similarities based on clip-level representations without additional optimization. We recognize the potential benefits of further optimization after obtaining the clip-level representation. Due to limited time, we conduct preliminary experiments by incorporating a parameter-free Text-Frame Attention Encoder in [1] to enhance video features (denoted as PFSim). With PFSim, TempMe achieved a **0.5%** improvement in R@1 and a **1.2%** increase in R-Sum, which validates your suggestion. These results will be included in the final version. Thank you for highlighting this valuable perspective, which we plan to investigate further in our future research.
>
> [1] Jin, P. et al., "DiffusionRet: Generative Text-Video Retrieval with Diffusion Model," ICCV, 2023.
>
> ### Table 1. Further optimization after backbone on MSRVTT with CLIP-ViT-32
> |Text-to-Video|R@1|R@5|R@10|R-Sum|
> |-|-|-|-|-|
> |TempMe|46.1|71.8|80.7|198.6|
> |TempMe+PFSim|**46.6**|**72.1**|**81.1**|**199.8**|
>
> **Question 2**: Lack of Memory Usage Comparison. Efficiency is not solely about GFLOPS—memory usage for training and inference is also crucial. While TempMe provides training acceleration, memory optimization would be even more beneficial. In Table 5, the authors report memory usage for CLIP-ViT-16, but it would be helpful to see detailed comparisons for CLIP-ViT-32 as well, particularly relative to previous methods.
>
> **Answer**: Thank you for raising this important point. In Table 2, we presents the memory useage during training and inference for CLIP-ViT-32. For all experiments, memory usage during training and inference are measured on 4 A100 GPUs, each processing a batch size of 32. These results will be included in the final version.
>
> (1) Although our TempMe achieves lower GFLOPs, it exhibits comparable inference memory usage to ToMe.  We speculate that this is partially influenced by the quadratic complexity of attention mechanisms, particularly the maximum token length processed (denoted as Attn Capacity). Specifically, ToMe’s Attn Capacity is 50 at layer 1, while our TempMe’s Attn Capacity is 118 at layer 11. Additionally, a similar trend in memory usage is observed in full fine-tuning with CLIP-ViT-16 in Table 5 of our revised paper. Despite comparable inference memory usage, our TempMe achieves a significant performance improvement, with a **3.2%** increase in R@1 and a **7.2%** increase in R-Sum, demonstrating its overall effectiveness.
>
> (2) Compared to previous PEFT TVR methods such as VoP and DGL, our TempMe requires significantly less memory during training and inference, validating the efficiency of our memory optimization. Specifically, Our TempMe not only improves **1.4**% R@1 but also achieves greater efficiency in terms of both model complexity and memory usage.
>
> ### Table 2. Memory usage comparisons on MSRVTT with CLIP-ViT-32
> |Text-to-Video|GFLOPs|Train Memory (MB)|Infer Memory (MB)|R@1|R-Sum|
> |-|-|-|-|-|-|
> |Prompt|58.2|16103|4941|40.3|184.0|
> |Adapter|53.1|15291|4785|41.9|190.2|
> |LoRA|53.0|16047|4783|43.7|193.0|
> |VoP|58.0|18913|6275|44.7|194.4|
> |DGL|67.4|18863|5203|44.6|194.8|
> |ToMe|40.2|13701|4737|42.9|191.4|
> |TempMe|**34.8**|**12381**|**4735**|**46.1**|**198.6**|

---

> > ### Author Response · Authors · 2024-11-26
> > **Official Comment by Authors**
> >
> > Thank you for your thoughtful comments and for raising your score.  We are glad our responses addressed your main concern about memory cost. As you suggested, we will incorporate the transformer version of DGL [1] (9.57M) in the final paper. We will revise our final paper accordingly based on your constructive comments.
> >
> > [1] Yang et al. "DGL: Dynamic Global-Local Prompt Tuning for Text-Video Retrieval." AAAI, 2024.

---

> ### Author Response · Authors · 2024-11-22
> **Author Response to Reviewer A4xM (2/3)**
>
> **Question 3**: Limited Novelty. The intra-clip merging process is highly similar to the original token merging approach [3], which limits the novelty of this method. In fact, the flowchart for intra-clip merging in Figure 3 closely resembles the diagram in Token Merging (ToMe), making it difficult to discern substantial differences. Essentially, the only distinct contribution is the cross-clip merging mechanism.
>
> **Answer**: Thank you for your feedback. (1) To clarify, ToMe and our TempMe address different domains and challenges. ToMe lacks mechanisms to reduce temporal redundancy or fine-tune temporal modeling across frames,  both of which are critical in text-video retrieval. In contrast, our innovative framework holistically integrates multiple proposed components, achieving state-of-the-art results in both accuracy and efficiency. (2) While we acknowledge that intra-clip merging is inspired by ToMe, its integration into the ClipMe Block serves a distinct and complementary purpose to novel cross-clip merging. As depicted in Figures 5b and 5c of our manuscript, intra-clip merging further optimizes token reduction, complementing cross-clip merging to enhance efficiency without sacrificing accuracy. Unlike ToMe, which focuses solely on spatial redundancy within a single image, our ClipMe Block combines cross-clip merging and intra-clip merging to enhance temporal modeling and reduce temporal redundancy.
>
> **Question 4.1**: Efficiency and Early Layer Merging. Performing token merging in earlier layers of CLIP could theoretically reduce both GFLOPS and memory usage, though it might harm performance metrics like R@1. TempMe, however, only applies merging in the last three layers with a 12-6-3-1 merging pattern.
>
> **Answer**: Thank you for your insightful comments. (1) As shown in Table 3, we agree that early layer merging can reduce GFLOPs and training memory usage. However, it may lead to higher inference memory usage due to limitations in Attn Capacity (see our answer to Question 2). Specifically, early layer merging’s Attn Capacity is 171 at layer 3, potentially increasing inference memory usage. (2) Additionally, Figure 5a in our manuscript shows that merging at earlier layers leads to noticeable performance degradation. This is likely because early layers in the CLIP image encoder capture low-level features, which are critical for frame-specific details but inadequate for modeling temporal relationships. In contrast, merging in later layers takes advantage of more semantic features that are better suited for learning spatio-temporal relationships.
>
> ### Table 3. Comparisons with early layer merging on MSRVTT with CLIP-ViT-32
> |Text-to-Video|GFLOPs|Train Memory (MB)|Infer Memory (MB)|R@1|R-Sum|
> |-|-|-|-|-|-|
> |$12\xrightarrow{1}6\xrightarrow{2}3\xrightarrow{3}1$|**13.7**|**7654**|4771|37.1|178.6|
> |$12\xrightarrow{9}6\xrightarrow{10}3\xrightarrow{11}1$|34.8|12381|**4735**|**46.1**|**198.6**|
>
> **Question 4.2**: A potential improvement could be to increase the input frames and match GFLOPS with previous PEFT methods, potentially enhancing R@1 without excessive memory usage. Additionally, it would be beneficial to provide memory usage details for CLIP-ViT-32 to understand whether further optimization is feasible.
>
> **Answer**: Thank you for your insightful suggestions. For fair comparisons, our TempMe is consistent with existing PEFT TVR methods in sampling 12 frames on MSRVTT in Table 2 of our manuscript. To explore the impact of longer input frames, we conduct additional experiments in Table 4. Increasing the input frames to 18 significantly improves performance by leveraging more temporal information. However, this comes at the cost of higher GFLOPs and memory usage, as each GPU needs to process 32x18 frames. Compared to previous PEFT TVR methods like VoP and DGL, our TempMe with 18 frames achieves a significant improvement of **3.2%** R@1 and **6.1%** R-sum under comparable inference memory usage. These results highlight the effectiveness of our TempMe in utilizing memory resources efficiently while achieving superior performance. These experiments will be included in the final version.
>
> ### Table 4. Effect of longer input frames on MSRVTT with CLIP-ViT-32
> |Text-to-Video|GFLOPs|Train Memory (MB)|Infer Memory (MB)|R@1|R-Sum|
> |-|-|-|-|-|-|
> |VoP|58.0|18913|6275|44.7|194.4|
> |DGL|67.4|18863|5203|44.6|194.8|
> |$12\xrightarrow{9}6\xrightarrow{10}3\xrightarrow{11}1$|**34.8**|**12381**|**4735**|46.1|198.6|
> |$18\xrightarrow{9}9\xrightarrow{10}3\xrightarrow{11}1$|51.5|16410|5879|**47.9**|**200.9**|

---

> ### Author Response · Authors · 2024-11-22
> **Author Response to Reviewer A4xM (3/3)**
>
> **Question 5**: Metric Suggestion. Presenting an overall sum@R can be unclear for readers. R@1 is generally a more meaningful metric than sum@R and would offer a clearer understanding of the model's performance.
>
> **Answer**: Thanks for the suggestion. We agree that R@1 directly indicates the model's top-1 retrieval accuracy. To improve clarity, we have included R@1 in all experimental tables in the Experiments section of our revised paper. Regarding the use of R-Sum, it is the sum of R@1, R@5, and R@10. Metrics like R@5 and R@10 are particularly valuable in practical applications where users may prefer having multiple retrieval options for further selection. Thus, R-Sum provides a comprehensive evaluation of the model's retrieval performance across different ranks.
>
> **Question 6**: If the memory usage of TempMe proves to be more efficient compared to previous PEFT or full finetuning methods, I would consider increasing my score.
>
> **Answer**: We sincerely appreciate your feedback and are glad to address your concern regarding memory usage. In Table 5, we evaluate TempMe against previous PEFT methods (see our answer to Question 2). Compared to previous PEFT TVR methods such as VoP and DGL, our TempMe requires significantly less memory during training and inference, validating the efficiency of our memory optimization. Specifically, Our TempMe achieves a significant **1.4%** R@1 improvement with only **65.6%** training memory usage. Additionally, in Table 5 of our revised paper, we present results for full fine-tuning. Compared to CLIP4Clip, TempMe achieves a significant **3.6**% R@1 improvement with only **75.2**% training memory usage. These results confirm that TempMe offers a more memory-efficient solution.
>
> ### Table 5. Memory usage comparisons of PEFT methods on MSRVTT with CLIP-ViT-32
> |Text-to-Video|GFLOPs|Train Memory (MB)|Infer Memory (MB)|R@1|R-Sum|
> |-|-|-|-|-|-|
> |Prompt|58.2|16103|4941|40.3|184.0|
> |Adapter|53.1|15291|4785|41.9|190.2|
> |LoRA|53.0|16047|4783|43.7|193.0|
> |VoP|58.0|18913|6275|44.7|194.4|
> |DGL|67.4|18863|5203|44.6|194.8|
> |ToMe|40.2|13701|4737|42.9|191.4|
> |TempMe|**34.8**|**12381**|**4735**|**46.1**|**198.6**|

---

> ### Author Response · Authors · 2024-11-25
> **Official Comment by Authors**
>
> We sincerely appreciate your time and effort in reviewing our paper. We would like to kindly inquire if our revisions and responses have addressed your concerns. Your further feedback would be invaluable.

---

> > ### Comment · Reviewer_A4xM · 2024-11-26
> >
> > Thank you to the authors for addressing my main concerns about memory cost. The 4000MB reduction in training memory and achieving R@1 ~48 with 18 frames with lower GLOPs than previous PEFL are impressive. Based on these improvements, I have raised my score to borderline accept.
> >
> > I would also like to remind the authors that DGL (ViT-32) achieves 45.8 R@1 with 9.57M trainable parameters, which is lower than VoP but performs better. Please revise Table 3 and include the new experiments, such as memory cost comparison and 18-frame results, in the final version.

---

### Official Review · Reviewer_AYTX · 2024-11-02

**Soundness:** 3
**Presentation:** 4
**Contribution:** 3
**Rating:** 6
**Confidence:** 3

**Summary:**

To enhance the efficiency of text-video retrieval, this paper introduces Temporal Token Merging (TempMe), a parameter- and inference-efficient architecture aimed at reducing spatial-temporal redundancy. The framework primarily consists of ImageMe Blocks for image merging and ClipMe Blocks for clip merging, achieving a progressive multi-granularity approach. Extensive experiments validate the superiority of the proposed method.

**Strengths:**

1, The paper addresses the issue of spatial-temporal redundancy in videos for text-video retrieval and introduces an efficient method that achieves faster training times and inference.
2, Extensive experiments and analyses of TempMe demonstrate its efficiency, effectiveness, and generalization capabilities.
3, The ClipMe Block mainly involves two steps, "Intra-clip Merging" and "Cross-clip Merging", each employing distinct methods for token grouping. This design effectively aids in information merging across clips.

**Weaknesses:**

1, Although the paper addresses spatial-temporal redundancy in text-video retrieval, there is already substantial work on token merging and pruning in video processing. This overlap may affect the perceived uniqueness of the proposed approach.
2, A few symbols lack adequate definitions, which may hinder readability. For instance, "R_c" in Section 3.2, while defined in Figure 3 as the ratio of kept tokens, should also be described in the text when first introduced for clarity.

**Questions:**

The authors should clarify how this work distinguishes itself from existing methods for token pruning or merging.

---

> ### Author Response · Authors · 2024-11-22
> **Author Response to Reviewer AYTX**
>
> Thank you very much for positive and thoughtful comments,  which have been instrumental in refining and enhancing the overall quality of our manuscript.
>
> **Question 1 & Question 3**:  (**Question 1**) Although the paper addresses spatial-temporal redundancy in text-video retrieval, there is already substantial work on token merging and pruning in video processing. This overlap may affect the perceived uniqueness of the proposed approach. (**Question 3**) The authors should clarify how this work distinguishes itself from existing methods for token pruning or merging.
>
> **Answer**:  Thank you for the insightful question. In this work, we focus on Text-Video Retrieval using pretrained CLIP, where each sampled frame is processed as an independent token set. Existing image/video token compression methods are limited to pruning or merging tokens within a single image/video token set,  without addressing token compression across multiple sets or incorporating temporal fine-tuning. In contrast, we have explored a practical and feasible path to reach both superior performance and computational efficiency. By fruitfully integrating parameter-efficient fine-tuning and token compression techniques, we propose TempMe and reach state-of-the-art performance. TempMe can progressively merge different frame token sets, and thus minimize spatio-temporal redundancy and enhance temporal modeling across frame. We have clarified this distinction in the Related Work section of our revised manuscript.
>
> **Question 2**: A few symbols lack adequate definitions, which may hinder readability. For instance, "R_c" in Section 3.2, while defined in Figure 3 as the ratio of kept tokens, should also be described in the text when first introduced for clarity.
>
> **Answer**: Thank you for pointing out this issue. In Section 3.2 of our revised paper, we have clarified the definition of $R_C$ and  $R_I$ by explicitly describing it in the text when it is first mentioned.

---

> > ### Comment · Reviewer_AYTX · 2024-11-25
> >
> > Thank you to the author for addressing my concerns and revising the previous issues. Considering that ICLR does not have a score of 7, and given that token merging and pruning as well as using pretrained CLIP are fairly common approaches in tasks like text-video retrieval and other video understanding tasks, the level of novelty is not sufficient to warrant a score of 8. Therefore, I will maintain my score of 6, which is already a positive rating.

---

> ### Author Response · Authors · 2024-11-25
> **Official Comment by Authors**
>
> Thank you for your acknowledgment of our efforts in addressing your concerns. Although the ICLR scoring system doesn’t include a 7, we’ll happily take your "implicit 7" as a sign of encouragement. We sincerely appreciate your positive and constructive evaluation.

---

### Author Response · Authors · 2024-11-22
**Author response to all reviewers**

We sincerely thank all reviewers for their valuable feedback and constructive comments. We have carefully addressed all comments and questions, with the updates in our revised paper highlighted in blue. Below, we summarize the key revisions and additional experiments conducted:

1. Following Reviewer AYTX's suggestions, we have clarified the distinctions between our method and existing token compression methods in the Related Work section.

2. Following Reviewer A4xM's suggestions, we conduct additional experiments to demonstrate the memory efficiency of our method. Furthermore, we validate that optimizing based on clip-level representations enhances performance. We also analyze the effect of merging positions and explore the effect of increasing the number of sampled frames. These results will be included in the final version of the paper.

3. Following Reviewer JLDW's suggestions, we have revised the Abstract and Introduction sections to enhance clarity and readability.

Thanks again for the reviewers' insightful feedback and their time spent reviewing our paper. We remain available for further comments or concerns.

---

### Meta-Review · Area_Chair_da7i · 2024-12-14

**Metareview:**

This paper proposed a new temporal token merging method for text-video retrieval to educe temporal redundancy and training/inference cost. The motivation is reasonable and the experiments are comprehensive, demonstrating efficiency, effectiveness, and generalization capabilities.  After rebuttal, the main weakness of this paper is that the level of novelty is not sufficient to warrant a score of 8. Since all reviewers leaned to accept this paper, I recommend accept to it.

**Additional Comments On Reviewer Discussion:**

The concerns raised by reviewers include insufficient experiments, quality of writing, lack of discussion of related works and limited performance improvement. The authors’ rebuttals addressed these concerns well, and two reviewers raised their ratings. The other two reviewers maintained their positive ratings.

---

### Decision · Program_Chairs · 2025-01-22

Accept (Poster)